# Type-Compliant Adaptation Cascades: Adapting Programmatic LM Workflows to Data

**Chu-Cheng Lin, Daiyi Peng, Yifeng Lu, Ming Zhang, Eugene Ie**
Google
{`kitsing,daiyip,yifenglu,mingzhang,eugeneie`}@google.com

## Abstract

Reliably composing Large Language Models (LLMs) for complex, multi-step workflows remains a significant challenge. The dominant paradigm — optimizing discrete prompts in a pipeline — is notoriously brittle and struggles to enforce the formal compliance required for structured tasks. We introduce Type-Compliant Adaptation Cascades (TACs), a framework that recasts workflow adaptation as learning typed probabilistic programs. TACs treat the entire workflow, which is composed of parameter-efficiently adapted LLMs and deterministic logic, as an unnormalized joint distribution. This enables principled, gradient-based training even with latent intermediate structures. We provide theoretical justification for our tractable optimization objective, proving that the optimization bias vanishes as the model learns type compliance. Empirically, TACs significantly outperform state-of-the-art prompt-optimization baselines. Gains are particularly pronounced on structured tasks, improving FinQA from 12.0% to 24.7% for a Qwen 3 8B model, MGSM-SymPy from 57.1% to 75.9% for a Gemma 2 27B model, MGSM from 1.6% to 27.3%, and MuSR from 36.5% to 62.6% for a Gemma 7B model. TACs offer a robust and theoretically grounded paradigm for developing reliable, task-compliant LLM systems.

## 1 Introduction

The expressive power of Large Language Models (LLMs) has catalyzed the rapid development of programmatically composed workflows and agentic systems (Khattab et al., 2022; Chase, 2022; Yao et al., 2023; Wu et al., 2024). By chaining model calls and integrating deterministic logic, practitioners construct complex systems capable of multi-step reasoning and interaction. However, the dominant paradigm for adapting these systems — optimizing discrete prompts within the pipeline — is notoriously brittle (Cao et al., 2024) and struggles to enforce the formal compliance required for structured tasks. Optimization often devolves into a difficult discrete search problem (Pryzant et al., 2023; Yuksekgonul et al., 2025), relying on heuristics that are computationally expensive and difficult to scale.

In this paper, we propose a fundamental shift in perspective: we recast the entire LLM workflow as a **typed probabilistic program**. Instead of optimizing the *inputs* (prompts) to a fixed system, we optimize the *program parameters*. We treat the workflow as a parametric latent variable model where each step is a probabilistic transformation backed by a parameter-efficient fine-tuning (PEFT) adaptor. This transforms workflow adaptation from an ad-hoc, discrete search problem into a principled, gradient-based optimization task focused on data likelihood maximization.

We introduce Type-Compliant Adaptation Cascades (TACs), a framework designed specifically for learning these typed, unnormalized probabilistic programs. TACs treat the entire workflow, composed of adapted LLMs and deterministic logic, as a joint unnormalized distribution. To enable tractable optimization, we propose the TACSTaR algorithm, a generalization of the Self-Taught Reasoner (STaR) (Zelikman et al., 2022) formalized within an MC-EM framework.

Crucially, we provide theoretical justification for optimizing the tractable unnormalized likelihood directly. We prove that the bias introduced by ignoring the partition function gradient is bounded by the degree of type violation. As the model learns to comply with the workflow's type constraints during training, the bias vanishes, and the optimization converges to the true maximum likelihood

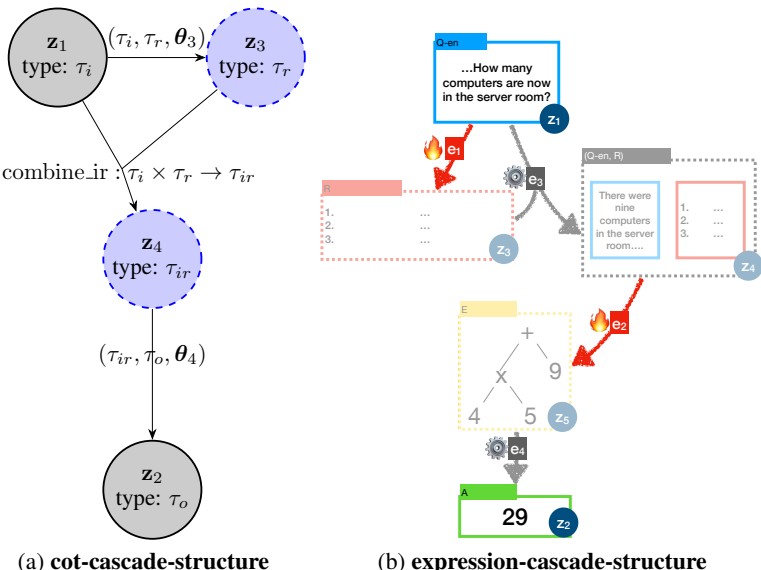

(a) **cot-cascade-structure**          (b) **expression-cascade-structure**

Figure 1: Two TAC workflow patterns experimented in this paper. We illustrate the more complicated Fig. 1b with example node values (we also explore additional patterns in §B). Dashed-boundary nodes indicate variables whose values are not available in annotated data, and solid-boundary nodes indicate nodes with training time observable values. A main message of this work is that **we can treat an entire typed workflow as a single probabilistic program, whose parameters are lightweight PEFT modules, allowing end-to-end training with latent variables**, instead of defining workflows imperatively as fixed-parameter systems.

solution (Theorems 1 and 2). Furthermore, this probabilistic framing allows us to decouple inference from training, enabling advanced techniques such as amortized inference to improve the E-step during optimization.

Our primary contributions are:

- **Framework**. We formalize typed LM workflows as *unnormalized probabilistic programs*, where type contracts restrict the support of learned transformations.
- **Theory**. We propose TACSTaR, a tractable optimization algorithm, and prove that its optimization bias vanishes as the model learns type compliance during training.
- **Practice**. Across reasoning-heavy tasks (MGSM, MGSM-SymPy, FinQA, MuSR) and model families (Gemma, Qwen), TACs consistently outperform strong DSPy prompt-optimization baselines. Gains are largest when (1) base models are smaller and (2) tasks require strict structure. For example, on MGSM-SymPy with a Gemma 27B model, TACs achieve $75.9$ vs. $57.1$; on FinQA, $34.0$ vs. $12.7$ (Gemma 27B) and $24.7$ vs. $12.0$ (Qwen 3 8B). With a Gemma 7B model, MGSM improves from $1.6$ to $27.3$, and MuSR from $36.5$ to $62.6$.

**Summary of results.**    (1) Gradient-based adaptation of typed probabilistic programs is markedly more effective and compute-efficient than discrete prompt search for structured tasks. (2) Flexible posterior inference, such as amortization, improves training stability and performance. (3) Empirically, the estimated type compliance mass $\mathcal{Z}_{\boldsymbol{\theta}}$ rises rapidly during training, supporting our theoretical justification for the unnormalized objective.

## 2    TYPE-COMPLIANT ADAPTOR CASCADES

The core idea of TACs is to decompose a task into a hypergraph of interconnected transformations. Formally, a TAC is represented as a directed acyclic hypergraph (DAH) $C = (\mathbf{Z}, \mathbf{E})$.[1] The acyclic

---

[1]We use a reasoning workflow that generates domain-specific code, illustrated in Fig. 1b, as a running example. The task is to take a math question in English (input type `Q_en`), generate a step-by-step rationale (intermediate type `R`), convert the rationale into a formal arithmetic expression (intermediate type `E`), and finally, have a

constraint ensures that the workflow has a well-defined topological order for execution and guarantees termination of the generative process.

**Nodes.** The nodes $\mathbf{Z} = \{\mathbf{z}_1, \mathbf{z}_2, \ldots, \mathbf{z}_M\}$ in a TAC act as containers for typed data. Each node $\mathbf{z}_m$ is associated with a specific data type $\tau \in \mathcal{T}$, and holds string representations $\in \Sigma^*$ for $\tau$-typed objects. Special nodes are designated as the **input node $\mathbf{z}_1$** and the **output node $\mathbf{z}_2$** (*e.g.*, holding the initial question of type Q_en and the final answer of type A in Fig. 1b, respectively).

**Hyperedges.** Hyperedges $\mathbf{E} = \{e_1, e_2, \ldots, e_K\}$ define the transformations between nodes. A hyperedge $e_k$ connects a set of source nodes $S_k \subseteq \mathbf{Z}$ (its inputs) to a set of target nodes $T_k \subseteq \mathbf{Z}$ (its outputs). Transformations in TACs can be either learnable (LM adaptors) or fixed (deterministic algorithms):

- **LM adaptor hyperedges.** These are stochastic transformations implemented by PEFT-adapted LMs. An adaptor $(\tau_i, \tau_o, \boldsymbol{\theta})$ defines an unnormalized distribution over $\boldsymbol{y} \in \Sigma^*$ given input string $\boldsymbol{x}$:[2]

$$\tilde{p}(\boldsymbol{y} \mid \boldsymbol{x}; \boldsymbol{\theta}) = p_{LM}(\boldsymbol{y} \mid \boldsymbol{x}; \boldsymbol{\theta}) \mathbb{I}(\mathbf{z}_t \in \text{valid}(\tau_o)), \tag{1}$$

  where $p_{LM}(\cdot \mid \boldsymbol{x}; \boldsymbol{\theta})$ is a normalized distribution over strings, conditioned on $\tau_i$-typed string representation $\boldsymbol{x}$, and parametrized by adaptor parameters $\boldsymbol{\theta}$, and $\text{valid}(\tau_o) \subseteq \Sigma^*$ is the set of strings that represent valid $\tau_o$-typed objects (we will further discuss them in §2.1).
- **Deterministic algorithm hyperedges.** These are fixed, non-learnable transformations, such as a self-contained Python function. A deterministic algorithm $f$ maps an input object of type $\tau_i$ to an output object of type $\tau_o$. Under the probabilistic view, we represent them as $\delta$ distributions:

$$\tilde{p}(\boldsymbol{y} \mid \boldsymbol{x}; f) = \delta_{\text{canon}(f(\text{parse}(x, \tau_i)))}(y) \tag{2}$$

  where canon (see §2.1) produces a canonicalized string for an object, and parse converts strings back to typed objects.

## 2.1 Interfacing LLMs with Typed Data: Parsing and Canonicalization

To integrate LLMs, which operate on strings ($\Sigma^*$), into a typed workflow, TACs require mechanisms to bridge the gap between strings and typed objects ($\mathcal{O}$). This bridge is typically handled by data validation libraries such as Pydantic[3] and PyGlove (Peng et al., 2020).[4] We formalize this conversion using two essential operations: parse and canon.

**Parsing (`parse`).** When an LM adaptor produces an output string $\boldsymbol{y}$ intended to represent an object of type $\tau_o$, this string is validated and converted into a usable typed object by the algorithm $\text{parse} : \Sigma^* \times \mathcal{T} \to \mathcal{O} \cup \{\text{error}\}$.[5] The expressivity of the TAC type system is determined by the implementation of parse and canon. In this work, we leverage Langfun, which supports primitive types, compound types (e.g., Python classes), and recursive data types (an example is listed in Listing 7).

For example, in Fig. 1b, $\mathbf{z}_5$ has the deterministic function $e_4$ as an outgoing edge. During execution of the probabilistic program, $\text{parse}(\mathbf{z}_5, \text{E})$ attempts to convert $\mathbf{z}_5$ into a SymPy expression object (typed E). If the conversion fails, an error is signaled. For convenience, we use $\text{valid}(\tau) = \{\text{parse}(\boldsymbol{y}, \tau) \neq \text{error} \mid \boldsymbol{y} \in \Sigma^*\}$ to denote valid string representations of $\tau$.

---

deterministic function evaluate this expression to produce the answer (output type A). This section formalizes how such an intuitive sketch is realized within the TAC framework.

[2]This distribution may be unnormalized because while $p_{LM}$ is a distribution over all strings, Eq. (1) restricts the support to only strings that are valid instances of $\tau_o$. Thus, the total probability mass may sum to less than 1 if the LM assigns probability to invalid strings.

[3]https://github.com/pydantic/pydantic

[4]Examples of type-specifying prompts generated by Langfun (which leverages PyGlove) are listed in §N. Langfun supports primitive types, compound types (*e.g.*, Pyglove/Pydantic objects), and recursive types (e.g., Expressions in Listing 7).

[5]We note that while primitive data types (*e.g.*, Python types str and list) appear in common workflows, parse can be any computable function, and can be leveraged by a practitioner to implement complex business logic. For example, one can define a Python custom type CoherentDialog where valid objects are strings deemed coherent by an external LLM-backed classifier, and adapt LM adaptors in a TAC to generate and work with such objects. Implementation details are further discussed in §E.

**Canonicalization (`canon`).** Conversely, inputs of LM adaptor hyperedges must be converted into a consistent string format that the adaptor expects. The `canon` : $\mathcal{O} \to \Sigma^*$ operation maps a typed object to a unique string representation — we call such strings *canonicalized*. The invertibility of `canon` (*i.e.*, `parse(canon`$(o), \tau_o) = o$) in turn ensures that deterministic hyperedges have support over only one string given a valid input, eliminating spurious ambiguity (Cohen et al., 2012).

## 2.2 TACS AS PROGRAMS AND DISTRIBUTIONS

TACs are finite-graph (DAH) unnormalized probabilistic programs where type validators act as hard support constraints. They thus admit both a program view, and also a distribution view[6]:

- **Program view.** Executing a TAC in the forward direction involves processing data through the hypergraph, respecting the topological order of nodes and hyperedges. Using our running example from Fig. 1b: the process traverses the hypergraph, starting at the input variable $\mathbf{z}_1$ (typed Q_en), and ending at the output variable $\mathbf{z}_2$ (typed A). A general process is described in Algorithm 1.
- **Distribution view.** TACs define unnormalized joint probability distributions over all node assignments $\mathbf{Z}^* = (\mathbf{z}_1^*, \mathbf{z}_2^*, \ldots, \mathbf{z}_M^*)$. This score reflects the plausibility of a complete execution trace according to the model's components:

$$\log \tilde{p}_{\boldsymbol{\theta}}(\mathbf{Z}^*) = \sum_k \log \tilde{p}_{\boldsymbol{\theta}}(\{\mathbf{z}_t^*\}_{t \in T_k} \mid \{\mathbf{z}_s^*\}_{s \in S_k}; e_k), \tag{3}$$

where $\boldsymbol{\theta}$ represents all adaptor parameters used in the TAC, and $\tilde{p}_{\boldsymbol{\theta}}(\cdot|\cdot; e_k)$ is the conditional probability defined by the LM adaptor (Eq. (1)) or deterministic algorithm (Eq. (2)) associated with $e_k$. The unnormalized distribution view connects TACs to the broader family of language model cascades (Dohan et al., 2022), but with the key distinction that TACs are designed for end-to-end adaptation.

## 3 ADAPTING TACS

The goal of adapting a TAC is to maximize the marginalized likelihood of the training data. Since TACs generally define distributions over unobserved (latent) intermediate variables, Monte Carlo Expectation-Maximization (MC-EM) algorithms (Wei & Tanner, 1990) provide a suitable training paradigm.[7]

However, adapting TACs presents a challenge. As TACs are generally unnormalized models, proper Maximum Likelihood Estimation (MLE) updates in the M-step require computing partition function gradients. Denoting the partition function summing all possible assignments as $\mathcal{Z}_{\boldsymbol{\theta}} = \sum_{\mathbf{Z}'} \tilde{p}_{\boldsymbol{\theta}}(\mathbf{Z}')$, the gradient of the log-likelihood $\mathcal{L}(\boldsymbol{\theta}) = \log p(\mathbf{Z}^*)$ is:

$$\nabla_{\boldsymbol{\theta}} \mathcal{L} = \nabla_{\boldsymbol{\theta}} \log \tilde{p}_{\boldsymbol{\theta}}(\mathbf{Z}^*) - \nabla_{\boldsymbol{\theta}} \log \mathcal{Z}_{\boldsymbol{\theta}}. \tag{4}$$

Estimation of the log partition function's gradients $\nabla_{\boldsymbol{\theta}} \log \mathcal{Z}_{\boldsymbol{\theta}}$ is typically intractable, expensive, and can have high variance (Goodfellow et al., 2016).

### 3.1 TRACTABLE OPTIMIZATION VIA COMPLIANCE

To overcome the intractable partition function gradient, we propose optimizing for the unnormalized log-likelihood $\mathcal{L}'(\boldsymbol{\theta}) = \log \tilde{p}_{\boldsymbol{\theta}}(\mathbf{Z}^*)$ instead, effectively dropping the $\nabla_{\boldsymbol{\theta}} \log \mathcal{Z}_{\boldsymbol{\theta}}$ term from Eq. (4).

While ignoring the partition function gradient generally leads to biased estimation, the TAC formalism ensures this strategy is both tractable and robust. This becomes evident as we rewrite $\mathcal{L}'(\boldsymbol{\theta}) = \mathcal{L}(\boldsymbol{\theta}) + \log \mathcal{Z}_{\boldsymbol{\theta}}$: optimizing the unnormalized likelihood $L'(\boldsymbol{\theta})$ is equivalent to jointly maximizing the normalized likelihood $\mathcal{L}(\boldsymbol{\theta})$ and the model's type compliance (the partition function $\log \mathcal{Z}_{\boldsymbol{\theta}}$ is maximized at $\log \mathcal{Z}_{\boldsymbol{\theta}} = 0$ when $\boldsymbol{\theta}$ is well-specified). We now provide theoretical justification for this approach, under the assumption that the adapted models can perfectly model type-valid outputs (*i.e.*, the model family is well-specified):[8]

---

[6]These two views are also summarized in Table 2.

[7]We acknowledge that another reasonable approach for training TACs is reinforcement learning, and note the connection between TACSTaR and RL in §A.

[8]We refer the reader to §D for proofs of formal statements in this section.

**Theorem 1.** *Let $\boldsymbol{\Theta}$ be the entire parameter space and let $\boldsymbol{\Theta}' \subseteq \boldsymbol{\Theta}$ be the subset of well-specified parameters. Assume $\boldsymbol{\theta}^*$ uniquely maximizes the normalized likelihood $p_{\boldsymbol{\theta}}(\mathbf{z}_{2..M}|\mathbf{z}_1)$ and resides $\in \boldsymbol{\Theta}'$. Then, $\hat{\boldsymbol{\theta}} = \arg\max_{\boldsymbol{\theta} \in \boldsymbol{\Theta}} \tilde{p}_{\boldsymbol{\theta}}(\mathbf{z}_{2..M}|\mathbf{z}_1) \implies \hat{\boldsymbol{\theta}} = \boldsymbol{\theta}^*$.*

Moreover, while optimizing $\mathcal{L}'(\boldsymbol{\theta})$ introduces a bias by ignoring the gradient term $\nabla_{\boldsymbol{\theta}} \log \mathcal{Z}_{\boldsymbol{\theta}}$, this bias is bounded below a constant multiplicative factor of $(1 - \mathcal{Z}_{\boldsymbol{\theta}})$ under the common assumption that $\|\nabla_{\boldsymbol{\theta}} p_{LM}(\cdot \mid \boldsymbol{x}; \boldsymbol{\theta})\|$ is uniformly bounded:

**Theorem 2.** *Let $\boldsymbol{\theta} = \{\boldsymbol{\theta}_1 \ldots \boldsymbol{\theta}_K\}$ be the union of a $K$-adaptor TAC's LM adaptor parameters . If $\forall \mathbf{z}_{k,1} \in \Sigma^*, \mathbf{z}_{k,2} \in \Sigma^*, \|\nabla \boldsymbol{\theta} \left(\sum \log p_{LM}(\mathbf{z}_{k,2} \mid \mathbf{z}_{k,1}; \boldsymbol{\theta})\right)\|_\infty \leq G$, then $\nabla_{\boldsymbol{\theta}} \log \mathcal{Z}_{\boldsymbol{\theta}} \leq 2G(1 - \mathcal{Z}_{\boldsymbol{\theta}})$.*

Theorems 1 and 2 provide theoretical assurance that if the model achieves high type compliance as we optimize for $\mathcal{L}'(\boldsymbol{\theta})$, the optimization bias vanishes, and the update approaches the true MLE update. Empirically, we observe that training rapidly drives $\mathcal{Z}_{\boldsymbol{\theta}}$ towards 1 (§4.4).

## 3.2   TACSTAR

We introduce the TACSTaR algorithm (Algorithm 3), which generalizes the Self-Taught Reasoner (STaR) algorithm (Zelikman et al., 2022) to the TAC framework. TACSTaR employs an iterative MC-EM approach to optimize the tractable objective $\mathcal{L}'(\boldsymbol{\theta})$ (§3.1). It alternates between E- and M-steps:

- **E-step: Sampling Latent Variables.** The E-step aims to sample complete, valid execution traces $\mathbf{Z}^*$ consistent with the training data. We first try to execute the TAC $C$ as a probabilistic program under the `forward` algorithm (Algorithm 1). If `forward` succeeds, we have a complete assignment of values $\mathbf{Z}^* = (\mathbf{z}_1^*, \mathbf{z}_2^*, \ldots, \mathbf{z}_M^*)$ and can proceed to M-step. Otherwise, we attempt a **rationalization heuristic** step. Inspired by the original STaR algorithm which conditions on the correct answer in the second attempt, we construct a 'fallback' TAC, whose input node takes $(x^*, y^*)$ as input, with the rest of the workflow unchanged. This essentially asks '*what intermediate steps would lead from $x^*$ to $y^*$?*', analogous to the inverse rendering problem (Ritchie et al., 2023). A forward pass is then executed on this new TAC to sample $(\mathbf{z}_2, \ldots, \mathbf{z}_M)$, now conditioned on both the original input $x^*$ and the desired output $y^*$. This encourages the generation of latent intermediate steps that are consistent with the correct final answer.
- **M-step: Parameter Optimization.** In the M-step, we update the adaptor parameters $\boldsymbol{\theta}$ by maximizing the unnormalized likelihood $\mathcal{L}'(\boldsymbol{\theta})$ of the samples collected in the E-step. [9]

## 3.3   AMORTIZED TACSTAR

The basic TACSTaR algorithm relies on a fixed 'fallback' heuristic during the E-step, which may be inefficient. Amortized TACSTaR (Algorithm 4) addresses this by generalizing the heuristic using parametric inference networks (Kingma & Welling, 2014; Mnih & Gregor, 2014), jointly trained to approximate the true posterior given observed input and outputs. By learning to propose better, task-adapted latent variable configurations, Amortized TACSTaR can lead to more efficient training and potentially better performance.

For model TAC $C$ with nodes $\mathbf{z}_1 \ldots \mathbf{z}_M$, we construct an inference network TAC $C'$ with nodes $\mathbf{z}_1' \ldots \mathbf{z}_M'$, which is trained alongside with $C$. In this work, we construct $\mathbf{z}_2' \ldots \mathbf{z}_M'$ to have the same types as $\mathbf{z}_2 \ldots \mathbf{z}_M$, except for its input node $\mathbf{z}_1'$, which has a type to represent the input-output pair $(x^*, y^*)$. Moreover, we construct $C'$ so that every adaptor hyperedge $e_k$ in $C$ has a counterpart $e_k'$ in $C'$ that is additionally conditioned on $\mathbf{z}_1'$. We train $C'$ alternately with $C$, with the goal of making the unnormalized distribution of $C'$ approximate the posterior over $C$'s intermediate nodes, conditioning on $(x^*, y^*)$ observations. Denoting the unnormalized distribution of $C'$ as $\tilde{q}_{\boldsymbol{\phi}}$ parametrized by adaptors' parameters $\boldsymbol{\phi}$, we hope to learn $\boldsymbol{\phi}$ such that $\tilde{q}_{\boldsymbol{\phi}}(\mathbf{z}_m' \mid \mathbf{z}_1' = \texttt{canon}((x^*, y^*))) \approx p_{\boldsymbol{\theta}}(\mathbf{z}_m \mid$

---

[9]**Remark on efficiency.** Since gradients of the log unnormalized probability decompose linearly as $\nabla_{\boldsymbol{\theta}} (\log \tilde{p}_{\boldsymbol{\theta}}(\mathbf{Z}^*)) = \sum_k \nabla_{\boldsymbol{\theta}} \log \tilde{p}_{\boldsymbol{\theta}}(\{\mathbf{z}_t^*\}_{t \in T_k} \mid \{\mathbf{z}_s^*\}_{s \in S_k}; e_k)$, computation of adaptors' gradients can be parallelized easily. This embarrassingly parallel structure ensures computational scalability, allowing the M-step to be efficiently distributed across available compute resources. Algorithm 2 computes $\log \tilde{p}_{\boldsymbol{\theta}}(\mathbf{Z}^*)$ and its gradients $\nabla_{\boldsymbol{\theta}} \log \tilde{p}_{\boldsymbol{\theta}}(\mathbf{Z}^*)$. These gradients are then used in a standard gradient-based optimization algorithm to update $\boldsymbol{\theta}$.

$\mathbf{z}_1 = x_c^*, \mathbf{z}_2 = y_c^*$), where $x_c^* = \texttt{canon}(x^*)$, $y_c^* = \texttt{canon}(y^*)$, $\forall m \in [2..M]$. Approximating the posterior $p_{\boldsymbol{\theta}}(\mathbf{z}_m \mid \mathbf{z}_1 = \texttt{canon}(x^*), \mathbf{z}_2 = \texttt{canon}(y^*))$ as $\hat{p}$ using self-normalized multiple importance sampling (Veach & Guibas, 1995), we optimize $\phi$ to minimize $\text{KL}[\hat{p}||q_{\phi}]$ following Bornschein & Bengio (2014); Lin & Eisner (2018). Empirically, we verify that the learned inference network $C'$ significantly reduces the KL divergence to the true posterior compared to the fixed fallback, confirming it provides a better approximation for training (§P).

## 4 EXPERIMENTS

To empirically validate TAC models, we conduct QA, code-like structured generation, and classification experiments on subsets of MGSM (Shi et al., 2023), FinQA (Chen et al., 2021), and MuSR (Sprague et al., 2024b) datasets,[10] adapting both instruction-tuned Gemma 7B and Gemma 2 27B (referred to as `gemma-1.1-7b-it` and `gemma-2-27b-it`) (Team et al., 2024), and Qwen 3 8B models (`Qwen3-8B`) (Yang et al., 2025). We aim to answer the following research questions:

- **(§4.2) Are TACs competitive against existing approaches?** TACs differ from existing LM adaptation approaches in two major ways: 1) TACs support gradient-based learning in a unified probabilistic programming framework (when compared against prior prompt optimization-focused LM programming frameworks such as DSPy); and 2) TACs support structured workflows by design (when compared to the original STaR algorithm). We hypothesize that such difference translates into meaningful performance improvements.
- **(§4.3) Is exploiting TACs' probabilistic flexibility effective?** Probability models (such as TACs) benefit from the decoupling of probabilistic modeling and inference procedures, allowing conditioning on additional observations *a posteriori*. We evaluate whether exploiting this flexibility is effective in two scenarios: 1) We compare Amortized TACSTaR (§3.3), which conditions on the output variable to learn a better proposal distribution for training, against the standard (unconditioned) TACSTaR; and 2) We evaluate TACs on a classification task, comparing the performance of unconstrained generation against a renormalized classifier that evaluates and normalizes the conditional probability of each possible output.
- **(§4.4) Does the model achieve high type compliance?** A key theoretical result (§3.2) is that the soundness and near-optimality of the TACSTaR optimization strategy rely on the model learning to comply with the workflow's type constraints (*i.e.*, driving the partition function $\mathcal{Z}_{\boldsymbol{\theta}} \to 1$). As type compliance increases, the gap between the tractable unnormalized likelihood and the true normalized likelihood ($\log \mathcal{Z}_{\boldsymbol{\theta}}$) closes. We estimate how $\mathcal{Z}_{\boldsymbol{\theta}}$ over TACSTaR epochs to verify that this gap is negligible after training.

### 4.1 EXPERIMENT SETUP

We provide an overview of our TAC and baseline DSPy setups below:

- **TACs.** We parametrize TAC adaptors to take the form of rank-1 LoRA models (Hu et al., 2022) on the attention weights, with $573, 440$; $1, 413, 120$; and $958, 464$ parameters per adaptor for `gemma-1.1-7b-it`, `gemma-2-27b-it` and `Qwen3-8B` respectively. For `parse` and `canon` implementations (§2.1), we leverage the LangFun library, which prompts LLMs to generate Python classes and objects, and parses their responses. LoRA weights are initialized ('zero-init') following Hu et al. (2022).
- **DSPy.** We conduct prompt-optimizing baseline experiments under DSPy, with base models served on vLLM. We subclass `dspy.Signature` to represent training examples, with property names and types identical to their TAC counterparts (some examples are listed in §G.2). We employ XGrammar (Dong et al., 2024) for schema-based constrained decoding for all experiments. We implement two types of reasoning workflows for all tasks: 1) the native `dspy.ChainOfThought` module, and 2) an explicitly two-step composite module that resembles **cot-cascade-structure** patterns under TACs. We experiment with various prompt optimization configurations under `dspy.MIPROv2` (Opsahl-Ong et al., 2024) and `dspy.BootstrapFewShotWithRandomSearch` (Khattab et al., 2024).

---

[10]We defer the study of how different TAC patterns affect performance to §B, where we expand our experiments to include HotPotQA tasks (Yang et al., 2018).

We conduct experiments on 5 reasoning-heavy tasks, on subsets from datasets MGSM[11] (Shi et al., 2023), FinQA (Yang et al., 2018), HotPotQA (Yang et al., 2018) and MuSR (Sprague et al., 2024b) respectively. Details of experiment setup are described in §G.

## 4.2 COMPARISON AGAINST PROMPT-OPTIMIZING AND UNTYPED STAR BASELINES.

Figure 2 lists MGSM, MGSM-SymPy, FinQA, and MuSR results from best-performing TACs and DSPy models. In addition, we compare the untyped (original) STaR against typed TAC results on MGSM on Gemma models.

**TACs are competitive against prompt-optimizing baseline methods.** We observe that TACs consistently and significantly outperform DSPy baselines in every setting. The performance gap is especially wide when 1) the base model is smaller, and 2) the task involves structured inputs (FinQA) or structured outputs (MGSM-SymPy).[12] We also discuss performance-computation cost trade-offs in §O; DSPy plateaus even at up to $\sim 9\times$ TAC inference tokens.

**No label leakage.** We emphasize that data validation (via Langfun) acts as a local function checking only the generated data against a type definition $\tau$. It does not access the ground truth label $y^*$ or input $x$, meaning label leakage is impossible by design. While structural constraints facilitate formatting, the massive performance gap between un-adapted and adapted models proves the model must still learn the complex reasoning required to arrive at the correct answer.

| Base Model | DSPy | TAC |
|---|---|---|
| gemma-1.1-7b-it | 0.7% | **9.7%** |
| gemma-2-27b-it | 12.7% | **34.0%** |
| Qwen3-8B | 12.0% | **24.7%** |

(a) FinQA

| Base Model | DSPy | TAC |
|---|---|---|
| gemma-1.1-7b-it | 36.5% | **62.6%** |
| gemma-2-27b-it | 51.5% | **65.0%** |
| Qwen3-8B | 61.5% | **63.7%** |

(b) MuSR

| Base Model | DSPy | TAC | STaR |
|---|---|---|---|
| gemma-1.1-7b-it | 1.6% | **27.3%** | 10.5% |
| gemma-2-27b-it | 81.9% | **82.2%** | 76.9% |

(c) MGSM

| Base Model | DSPy | TAC |
|---|---|---|
| gemma-2-27b-it | 57.1% | **75.9%** |

(d) MGSM-SymPy

Figure 2: Comparison between best performing prompt-optimizing methods under DSPy and TACs (full results can be found in Sections H to L). We report the best DSPy result for each task.

**Type compliance helps: TACSTaR compares favorably against the unstructured STaR baseline.** On the MGSM task (Fig. 2c), the original (untyped) STaR algorithm scored an average accuracy of 76.9 and 10.5 (from gemma-2-27b-it and gemma-1.1-7b-it respectively), lower than variants of reasoning TAC patterns on the same dataset. This demonstrates that the structured, typed approach of TACs improves performance over the untyped STaR baseline.

## 4.3 FLEXIBLE POSTERIOR INFERENCE HELPS TAC PERFORMANCE.

**Amortized inference at training time is effective.** The Amortized TACSTaR algorithm (§3.3) brings consistent improvement over vanilla TACSTaR on 3 tasks (Fig. 4a). Notably, the gains are most substantial on FinQA (+5.7 points). This suggests that amortized inference is particularly valuable for complex tasks where the initial sampling or fixed rationalization heuristics struggle to find valid latent traces, allowing the model to learn a more effective inference strategy.

---

[11]The MGSM-SymPy task uses the same problems of MGSM, but additionally restrict the outputs to be rational expressions under SymPy. This variant was specifically included to test the framework's ability to generate and comply with highly structured, code-like output.

[12]We also compare between TACSTaR-adapted and un-adapted models on the same LangFun prompts in §B.2, and find that TACSTaR consistently outperforms the un-adapted counterparts.

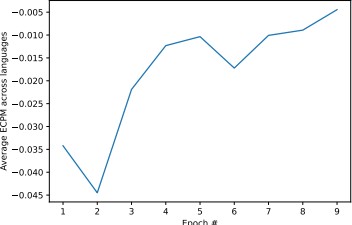

(a) Average estimate $\log \mathcal{Z}_{\theta}$ over validation set inputs versus # of TACSTaR epochs over MGSM languages. Note that later epochs (as early as epoch 5) do not have samples from all languages, as some languages early-stopped.

| At the end of epoch | Failure rate |
|---|---|
| 1 | 83.0% |
| 2 | 1.0% |
| 3 | 1.6% |
| 4 | 0.4% |

(b) Average MGSM training data parsing failure rate vs # of epochs of TACSTaR on `gemma-1.1-7b-it`. The pattern is **cot-cascade-structure**.

Figure 3: Type compliance during TAC training.

| Task | TACSTaR | Amortized TACSTaR |
|---|---|---|
| MGSM | 82.2 | 82.4 |
| FinQA | 36.0 | 41.7 |
| HotPotQA | 32.0 | 34.0 |

(a) Comparison between TACSTaR and Amortized TACSTaR on **cot-cascade-structure** / `gemma-2-27b-it`.

| Base Model | Cla. | Gen. |
|---|---|---|
| `gemma-1.1-7b-it` | 62.6 | 62.1 |
| `gemma-2-27b-it` | 65.0 | 51.6 |

(b) Comparison between classification and unconstrained generation results on MuSR.

Figure 4: Comparison between default and more informative inference methods.

**Classification with renormalized posterior at inference time is effective.** We leverage the probabilistic nature of TACs to estimate the output label posterior $p_{\theta}(\mathbf{z}_2 \mid \mathbf{z}_1)$ for the MuSR classification task. We achieve this by first estimating the unnormalized probability $\tilde{p}$ for each label using importance sampling, and then renormalizing these estimates over the finite label space (Self-Normalized Importance Sampling). The detailed formulation is described in §R. We output the label with the highest estimated probability. Figure 4b shows that the renormalized-posterior classifier outperforms unconstrained generation on both `gemma-1.1-7b-it` and `gemma-2-27b-it`.

**Inference network training in the Amortized TACSTaR algorithm is effective.** To quantify the effectiveness of self-normalized importance sampling (SNIS) under the trained inference networks in the Amortized TACSTaR algorithm, we conducted additional experiments with the `cot-cascade-structure` pattern to estimate reverse KL divergence, $\mathrm{KL}[\tilde{q}_{\phi}||p_{\theta}]$, which measures how well self-normalized importance samples from adapted proposal $q_{\phi}(\cdot \mid \mathbf{z}_1^*, \mathbf{z}_2^*)$ — as the mixture $\tilde{q}_{\phi}$ — approximates the true posterior $p_{\theta}(\cdot \mid \mathbf{z}_1^*, \mathbf{z}_2^*)$: $\mathrm{KL}[\tilde{q}_{\phi}||p_{\theta}] = 0$ when $\tilde{q}_{\phi} = p_{\theta}$. Since the partition function of $\sum_{\mathbf{z}} p_{\theta}(\mathbf{z}_3 = \mathbf{z}, \mathbf{z}_2 = \mathbf{z}_2^* \mid \mathbf{z}_1^*)$ is intractable, we estimate the difference between KL divergences, namely $\mathrm{KL}[\tilde{q}_{\phi}||p_{\theta}] - \mathrm{KL}[\tilde{p}_{\mathrm{fallback}}||p_{\theta}]$, comparing against SNIS distributions under the unamortized TACSTaR fallback.[13] We report the average difference over validation datasets on the latest MGSM checkpoints in Table 1. On average, SNIS distribution under the adapted $\tilde{q}_{\phi}$ is closer to the true posterior distribution $p_{\theta}(\cdot \mid \mathbf{z}_1^*, \mathbf{z}_2^*)$ than the counterpart under the default $p_{\mathrm{fallback}}$.

### 4.4 TAC MODELS RAPIDLY ACHIEVE HIGH TYPE COMPLIANCE.

We argued in §3.2 that optimizing the unnormalized likelihood drives the model towards structural compliance. The average MGSM parsing error rate during training (Fig. 3b) suggests that TACs learn compliance fast. We further empirically verify this by estimating the partition function $\mathcal{Z}_{\theta}$ — which represents the total probability mass the model assigns to type-compliant outputs (the Estimated Compliant Probability Mass, ECPM) — throughout training. We estimate $\log \mathcal{Z}_{\theta}$ on the validation sets of the MGSM benchmark during training of the **cot-cascade-structure** pattern on `gemma-1.1-7b-it`. We sample 100 generations of entire traces without type-compliant masking

---

[13]We derive a Monte Carlo estimation of this quantity in §P.

| Language | $\text{KL}[\tilde{q}_{\boldsymbol{\phi}}||p_{\boldsymbol{\theta}}] - \text{KL}[\tilde{p}_{\text{fallback}}||p_{\boldsymbol{\theta}}]$ |
|---|---|
| es | -0.71 |
| en | -3.72 |
| de | -0.79 |
| fr | -1.62 |
| zh | 3.07 |
| ja | 0.19 |
| ru | -1.42 |
| te | -2.61 |
| th | -0.05 |
| Avg. of all languages | -0.85 |

Table 1: Average KL divergence differences from the true posterior, between SNIS distributions under adapted and un-adapted fallback distributions. A negative value implies $\text{KL}[\tilde{q}_{\boldsymbol{\phi}}||p_{\boldsymbol{\theta}}] < \text{KL}[\tilde{p}_{\text{fallback}}||p_{\boldsymbol{\theta}}]$.

per input with temperature $= 1$, top-p $= 1$, and top-k set to the vocabulary size. Figure 3a shows that the model rapidly learns to comply with the type constraints.[14] The average $\log \mathcal{Z}_{\boldsymbol{\theta}}$ approaches $-0.005$ by epoch 9, corresponding to an ECPM of $\exp(-0.005) \approx 99.5\%$, and thus confirms that the degree of misspecification $(1 - \mathcal{Z}_{\boldsymbol{\theta}})$ is negligible. Since the difference between unnormalized and normalized likelihood gradients is bounded by a multiplicative factor of $(1 - \mathcal{Z}_{\boldsymbol{\theta}})$ (Theorem 2), our empirical estimates imply that the difference is indeed small at the end of training, and TACSTaR M-step (§3.2) approaches the true MLE update. Moreover, since $\log \mathcal{Z}_{\boldsymbol{\theta}}$ is the difference between normalized and unnormalized likelihoods, the small magnitude suggests it is practical to do model selection with unnormalized likelihood directly, after a few epochs of training.

## 5 CONCLUSION

We have presented Type-Compliant Adaptation Cascades (TACs), a novel probabilistic programming framework designed to empower ML practitioners to design trainable workflows that adapt to data. Our findings demonstrate that TACs' gradient-based learning paradigm is highly effective, consistently outperforming strong prompt-optimization baselines. Moreover, we also find flexible posterior inference of TACs at both training and inference time help with performance. We also find that empirically, the model learns to comply with type constraints fast in training, justifying the assumptions in our theoretical results. These results underscore the versatility and efficacy of TACs as a scalable paradigm for adapting to complex, reasoning-heavy tasks.

## ACKNOWLEDGMENTS

We thank Arya McCarthy, Aurora Wei, Debangshu Banerjee, Eric Li, Yaohui Guo, Yaoyiran Li, and the anonymous reviewers for their constructive comments and suggestions.

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

## APPENDICES

| Program View | Probabilistic View |
|---|---|
| $\tau$-typed object | Random variable $\in \Sigma^*$ restricted to strings $\in \text{valid}(\tau)$ |
| LM adaptor with weights $\boldsymbol{\theta}$, with output restricted to $\tau$-typed objects | Unnormalized conditional distribution $p_{LM}(\mathbf{z}_t \mid \mathbf{z}_s; \boldsymbol{\theta}) \mathbb{I}(\mathbf{z}_t \in \text{valid}(\tau))$ |
| Deterministic algorithm $f : \tau_i \to \tau_o$ | Degenerate distribution $\delta_{\texttt{canon}(f(\texttt{parse}(x, \tau_i)))}(y)$ |
| `parse` and `canon` functions that convert typed objects to/from LM inputs/outputs | Measurable maps between object domain $\mathcal{O}$ and string domain $\Sigma^*$ |
| Executing a workflow to obtain $\mathbf{z}_{1\ldots M}$ | Sampling from joint unnormalized probability $\tilde{p}_{\boldsymbol{\theta}}(\mathbf{z}_{1\ldots M}) = \prod_k \tilde{p}_{\boldsymbol{\theta}}(\mathbf{z}_{T_k} \mid \mathbf{z}_{S_k})$ |
| Probability that a stochastic workflow succeeds | $\mathcal{Z}_{\boldsymbol{\theta}} = \text{Pr}_{p_{\boldsymbol{\theta}}}(\text{all nodes are valid})$ |

Table 2: Dual semantics: how TAC concepts map between their program and probabilistic views.

## A  BACKGROUND AND RELATED WORK

TACs sit at the intersection of probabilistic modeling, workflow composition, and LLM adaptation. We organize the related work thematically.

**Formalizing and Executing LM Workflows.**  We approach LLM workflows from the perspective of probabilistic programming. Probabilistic programming languages (PPLs) tailored for machine learning, such as Edward (Tran et al., 2017) and Pyro (Bingham et al., 2019), combine differentiable components with stochastic control flow to define complex distributions. Notably, similar to First-Order PPLs (van de Meent et al., 2021), TACs specify finite-graph probabilistic models. However, TACs address a distinct challenge inherent to typed LLM workflows: enforcing type constraints restricts the LLM's support (via `parse`), rendering the distribution *unnormalized* ($\mathcal{Z}_{\boldsymbol{\theta}} \neq 1$). While standard PPLs perform posterior inference conditioning on observations (the `observe` statement), TACs require conditioning on both type compliance and observations. This allows us to retain LLM expressivity while enforcing validity, at the cost of a partition function $\mathcal{Z}_{\boldsymbol{\theta}} \neq 1$. TACs draw inspiration from classical structured prediction (Lafferty et al., 2001; Belanger & McCallum, 2016), which provides tools for handling unnormalized models. Our formulation connects these threads, treating type compliance itself as the partition function, enabling a specialized, tractable optimization objective (TACSTaR).

In contrast, programmatic LM frameworks such as DSPy (Khattab et al., 2022; 2024), LMQL (Beurer-Kellner et al., 2023), and LangChain (Chase, 2022) expose LMs through pipelines with declarative constraints. These systems typically optimize the *inputs* (prompts or few-shot exemplars) to a fixed system, rather than casting the entire workflow as a single probabilistic object with learnable continuous parameters and a likelihood objective. While some proposals optimize weights within such pipelines (e.g., `BetterTogether` (Soylu et al., 2024)), TACs differ fundamentally in their principled probabilistic formulation, enabling theoretically justified training (§3.2) and advanced inference techniques (§3.3).

**Optimizing Composed Systems.**  The standard approach to optimizing LM workflows involves difficult discrete optimization over the space of possible prompts, often addressed through heuristic search (Zhou et al., 2023; Pryzant et al., 2023; Yuksekgonul et al., 2025) or reinforcement learning (Jafari et al., 2024), both of which can suffer from high variance and computational cost.

TACs instead leverage gradient-based optimization. This builds upon methods that adapt LMs for reasoning, such as STaR (Zelikman et al., 2022) and ReFT (Trung et al., 2024), which were inspired by techniques like Chain-of-Thought (CoT) (Wei et al., 2022) and Self-Refine (Madaan et al., 2023). We adopt the spirit of STaR, but generalize it within a hypergraph framework to propose typed, multi-step rationalizations (§3.2). Furthermore, we introduce an amortized variant that learns to propose rationalizations, rather than relying solely on heuristics (§3.3).

**Enforcing Structure and Compliance.** To improve output reliability, various methods enforce grammar-based constraints during LLM generation (Poesia et al., 2022; Geng et al., 2023; McCarthy et al., 2023; Beurer-Kellner et al., 2024; Geng et al., 2025). These methods generally modify *local* conditional distributions over next tokens at inference time, masking out continuations incompatible with the grammar. In contrast, our objective learns parameters so that type-compliant trajectories carry increasing probability mass *globally*, improving both validity and task accuracy through training.

**Parameter-efficient adaptation.** LoRA and related PEFT methods (Houlsby et al., 2019; Hu et al., 2022; Li & Liang, 2021; Lester et al., 2021; Liu et al., 2022) enable light-weight adaptation. We use small adaptors to highlight data-efficiency and show that gains stem from *typed workflow learning* rather than sheer capacity.

**Connection to Reinforcement Learning.** The TACSTaR training procedure (§3.2) can also be viewed through the lens of policy optimization. As Zelikman et al. (2022) observed, the STaR objective closely resembles the REINFORCE algorithm (Williams, 1992). The M-step in TACSTaR can be interpreted as optimizing the workflow policy under REINFORCE with a binary reward for generating the correct output.

We adopt the MC-EM framing as it provides a principled approach for likelihood maximization in the presence of annotated output data. While advanced RL techniques (*e.g.*, PPO (Schulman et al., 2017) or actor-critic methods (Konda & Tsitsiklis, 1999)) might be applicable, they introduce complexity, such as training value functions, which are difficult to estimate over complex, typed latent spaces. Furthermore, the exploration challenge in sparse reward settings is significantly mitigated by the rationalization heuristic and the inference network in Amortized TACSTaR (§3.3), which guide sampling towards successful trajectories using known outputs.

# B  ADDITIONAL STUDIES ON WORKFLOW PATTERN DESIGN

In this section, we conduct additional experiments that vary the pattern structures, and evaluate how such changes affect performance. Specifically, we would like to answer the following questions:

- **(§B.2) Is adaptation with reasoning workflows effective?** The TAC framework gives practitioners great freedom in designing a workflow that reason in the process. We hypothesize that adapting with such explicit structures improves performance on tasks that require complex reasoning.
- **(§B.3) How do TAC design variations affect performance?** We evaluate how such TAC design variations for the same task affect performance.

## B.1  END-TO-END TRAINABLE WORKFLOWS AS TACS.

The declarative and flexible nature of TACs enable practitioners to rapidly implement end-to-end trainable workflows. We implement some common patterns as TACs:

- **Direct adaptation** of an LM to the downstream task without any latent structure corresponds to common supervised PEFT methods surveyed in §A. The **direct** pattern (Fig. 5a) is a singleton TAC with no latent nodes.
- **Adapting with latent rationales** corresponds to patterns that learn to generate rationales for the task at hand Zelikman et al. (2022). There are several possible TAC structure designs that incorporate rationales: for example, **cot-type-structure** (Fig. 5b) maps the input to a rationale-output typed object, from which the task output is deterministically extracted. Alternatively, **cot-cascade-structure** (Fig. 1a) introduce rationales as distinct nodes in the TAC hypergraph, which transforms into the task output under an adaptor.
- **Trainable self-refinement** refers to an end-to-end trainable variant of self-refine (Madaan et al., 2023), where the model first sketches a task output, and iteratively refines it. Without TAC, a practitioner would have to resort to manually writing tedious postprocessing functions for the intermediate results. On the other hand, the TAC counterpart **refine-structure** (Fig. 6 in §F) is straightforward.

For the MGSM-SymPy task, we experiment with the **expression-cascade-structure** pattern (Fig. 1b), which additionally imposes the constraint that the output must be a rational number represented by an

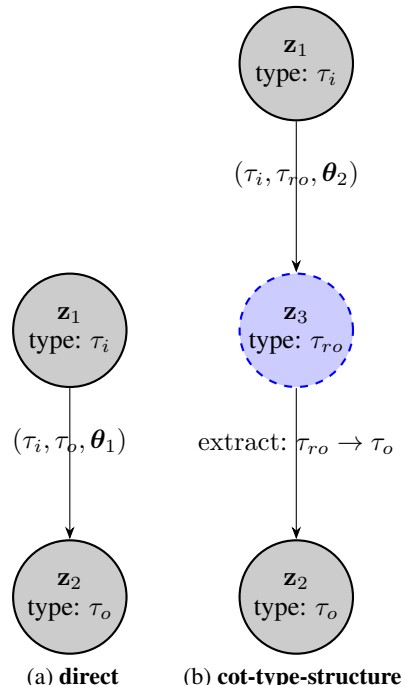

Figure 5: Workflow patterns experimented in this paper, with increasing structural complexity from left to right. In the most complicated pattern **expression-cascade-structure** we illustrate the workflow with example node values. Dashed-boundary nodes indicate variables that are not observed at training time. And solid-boundary nodes indicate nodes with training time observable values. A main message of this work is that instead of defining workflows imperatively as fixed-parameter systems, **we treat an entire typed workflow as a single probabilistic program, whose parameters are lightweight PEFT modules, allowing end-to-end training with latent variables.**

arithmetic expression tree. Such type constraints often reflect business logic (for example, we expect the MGSM dataset to have rational number answers), and may be necessary when the TAC forms a component in a larger system.

## B.2 EFFECTIVENESS OF ADAPTATION WITH REASONING WORKFLOWS

To evaluate whether adaptation with reasoning workflows is effective, we compare **cot-cascade-structure**, and **refine-structure** TACs against **direct** on the 3 tasks MGSM, FinQA and HotPotQA, on base models gemma-2-27b-it and gemma-1.1-7b-it. Table 3 shows that both **cot-cascade-structure** significantly outperforms **direct** on MGSM and FinQA on both gemma-2-27b-it and gemma-1.1-7b-it. But **cot-cascade-structure** slightly underperforms **direct** on HotPotQA. These results largely agree with the meta study done by Sprague et al. (2024a), which also reported that tasks that require arithmetic and symbolic reasoning, such as MGSM and FinQA, benefit the most from CoT, while a huge portion of previous work saw that CoT degrades performance for multihop QA. However, we note that the **refine-structure** TAC (Fig. 6) consistently outperform the **direct** baseline in all 3 tasks on gemma-2-27b-it, showcasing the effectiveness of the adaptive refinement paradigm.

**Task adaptation with TACSTaR is effective.** To evaluate whether the efficacy of TACs can be attributed to our proposed TACSTaR method, we also compare adapted TAC workflows against those with the same hypergraph structure, but with un-adapted weights (*i.e.*, all adaptors in the TAC use base model weights). Both TACSTaR trained and un-adapted models use the same structured Langfun prompts that are similar to examples listed in §N. The significant gap between adapted and un-adapted results in Table 4 indicate that the TACSTaR algorithm is effective. Notably, un-adapted models still

| Dataset | gemma-2-27b-it | | | gemma-1.1-7b-it | |
|---|---|---|---|---|---|
| | direct | cot-cascade-structure | refine-structure | direct | cot-cascade-structure |
| MGSM | 24.7 | **82.2** | 78.6 | 5.1 | **27.3** |
| FinQA | 17.3 | **36.0** | 23.7 | 3.0 | **9.7** |
| HotPotQA | 34.0 | 32.0 | **39.0** | — | — |

Table 3: Comparison between **direct** and reasoning workflows. For the MGSM dataset, we report per-language accuracies in Table 6. The difference between best performing runs and **direct** are statistically significant/marginally significant: for MGSM and FinQA $p < 0.05$ (both gemma-2-27b-it and gemma-1.1-7b-it), and for HotPotQA $p = 0.07$ under paired permutation tests. Per-language accuracy numbers of the MGSM dataset are in §H.

outperform **direct** workflows (listed in Table 3), indicating that Langfun's type-inducing prompts can invoke somewhat effective test-time computation over the TAC hypergraph structure.

| Task | Structure | TACSTaR | Un-adapted |
|---|---|---|---|
| MGSM | cot-cascade-structure | 82.2 | 45.4 |
| MGSM | cot-type-structure | 80.4 | 74.7 |
| MGSM-SymPy | expression-cascade-structure | 75.9 | 69.5 |
| FinQA | cot-cascade-structure | 36.0 | 13.0 |
| HotPotQA | refine-structure | 39.0 | 24.0 |

Table 4: Comparison between TACSTaR-adapted and un-adapted gemma-2-27b-it. The differences are all statistically significant ($p < 0.05$) under paired permutation tests.

### B.3 EFFECTS OF DIFFERENT TAC DESIGNS

**Decoupling rationale and output modeling helps performance.** **cot-cascade-structure** (Fig. 1a) achieves a higher score than **cot-type-structure** (Fig. 5b) on the MGSM task (Table 5), suggesting that modeling the rationale and task output generation with distinct adaptors helps performance. By using distinct adaptors, the workflow allows specialization: the first adaptor focuses on reasoning, while the second specializes in synthesis, reducing the complexity burden on a single monolithic step. The positive result again highlights how the TAC formalism can help practitioners iterate and experiment with different multi-adaptor cascade designs, which would be tedious otherwise.

**Robustness to Semantic Constraints.** Comparing performance on MGSM and the more constrained MGSM-SymPy task reveals a key advantage of the TAC framework's robustness. As shown in Table 5, the best-performing TAC model sees a modest performance drop, from 82.2% on MGSM to 75.9% on MGSM-SymPy, when required to generate a valid symbolic expression.[15] This contrasts sharply with the prompt-optimizing baseline (Fig. 2). The best DSPy configuration experiences a much more significant degradation, plummeting from 81.9% on MGSM to just 57.1% on MGSM-SymPy. The substantially smaller performance drop for TACs underscores the brittleness of discrete prompt optimization when faced with strict structural requirements. The TAC framework's gradient-based adaptation within a typed system proves to be significantly more resilient, making it a more reliable paradigm for tasks demanding structural compliance.

## C ALGORITHMS

### C.1 FORWARD AND BACKWARD

Algorithm 1 (forward) executes the probabilistic program represented by a TAC $C = (\mathbf{Z}, \mathbf{E})$. Starting from a given input node value $\mathbf{z}_1^*$, the algorithm traverses the hypergraph following a

---

[15]Sample expressions generated under **expression-cascade-structure** are listed in §M.

|  | MGSM | MGSM-SymPy |
|---|---|---|
| **cot-type-structure** | **cot-cascade-structure** | **expression-cascade-structure** |
| 80.4 | 82.2 | 75.9 |

Table 5: Effects of different TAC designs on the MGSM dataset, demonstrating the impact of workflow structure on performance. The **cot-cascade-structure** (which decouples rationale generation from the final answer synthesis) outperforms the monolithic **cot-type-structure**. The **expression-cascade-structure** result shows strong performance on the more constrained MGSM-SymPy task.

topological order, and terminates when all nodes $\in \mathbf{Z}$ have been visited. `forward` takes $C$ and $\mathbf{z}_1^*$ as input arguments. `forward` also takes the following as arguments:

- sampler configuration $\kappa$ for different sampling techniques, *e.g.*, varying temperature, nucleus, and top-$k$ sampling
- maximum number of sampling attempts

---

**Algorithm 1** TAC Forward Algorithm (`forward`)

---

**Input:** TAC cascade $C = (\mathbf{Z}, \mathbf{E})$ where $\mathbf{Z} = \{\mathbf{z}_1 \ldots \mathbf{z}_M\}$ and $\mathbf{E} = \{\mathbf{e}_1 \ldots \mathbf{e}_K\}$, input object: $\mathbf{z}_1^*$, sampler configuration $\kappa$, $N_{\max}$ for maximum number of sampling attempts.
**Output:** Sampled values $(\mathbf{z}_2^*, \ldots, \mathbf{z}_M^*)$.
  1: Determine a topological ordering of edges in $\mathbf{E}$. Let the sorted hyperedges be $e_1' \ldots e_K'$.
  2: $\mathbf{Z}_{\text{already\_sampled}}^* \leftarrow \{\mathbf{z}_1^*\}$.
  3: **for** $k \in [1..K]$ **do**
  4:     Assert the source nodes of $e_k'$ are a subset of $\mathbf{Z}_{\text{already\_sampled}}^*$.
  5:     **if** $e_k' = (\tau_i, \tau_o, \boldsymbol{\theta})$ is a type-constrained LM adaptor **then**
  6:         # type-constrained LM adaptors have a single source node and a single target node.
  7:         $\boldsymbol{x} \leftarrow$ canonicalized representation of $e_k'$'s source node.
  8:         **while** number of attempts $\leq N_{\max}$ **do**
  9:             Try draw $\boldsymbol{y} \sim p_{LM}(\cdot \mid \boldsymbol{x}; \boldsymbol{\theta}, \kappa)$
 10:             **if** `parse`$(\boldsymbol{y}, \tau_o) \neq$ error **then**
 11:                 $t \leftarrow$ index of $e_k'$'s target node.
 12:                 $\mathbf{z}_t^* \leftarrow \boldsymbol{y}$
 13:                 $\mathbf{Z}_{\text{already\_sampled}}^* \leftarrow \mathbf{Z}_{\text{already\_sampled}}^* \cup \{\mathbf{z}_t^*\}$
 14:                 break
 15:             **end if**
 16:         **end while**
 17:     **else if** $e_k'$ is a deterministic algorithm $f$ **then**
 18:         # **In this work we assume** $f$**'s inputs and outputs are sorted by node index in** $C$**.**
 19:         $\mathbf{O}_{f\text{input}} \leftarrow$ parsed objects of $e_k'$'s source nodes, sorted by node index.
 20:         $\mathbf{O}_{f\text{output}} \leftarrow f(\mathbf{O}_{f\text{input}})$
 21:         $\mathbf{Z}_{f\text{output}}^* \leftarrow$ canonicalized representations of objects $\in \mathbf{O}_{f\text{output}}$, sorted by node index.
 22:         $\mathbf{Z}_{\text{already\_sampled}}^* \leftarrow \mathbf{Z}_{\text{already\_sampled}}^* \cup \mathbf{Z}_{f\text{output}}^*$
 23:     **end if**
 24: **end for**
 25: **return** $\mathbf{Z}_{\text{already\_sampled}}^* - \{\mathbf{z}_1^*\}$.

---

Algorithm 2 (`backward`) takes as input $(C, \mathbf{Z}^*)$, where $C = (\mathbf{Z}, \mathbf{E})$ where $\mathbf{E} = (e_1 \ldots e_K)$ is a TAC, and $\mathbf{Z}^*$ are value assignments of $\mathbf{Z}$. We assume the log probability $p_{LM}(\boldsymbol{y} \mid \boldsymbol{x}; \boldsymbol{\theta}_k)$ is auto-differentiable with regard to all adaptor hyperedges in a TAC. Algorithm 2 returns unnormalized log joint probabilities of $\mathbf{Z}^*$ under $C$: $\log \tilde{p}_{\boldsymbol{\theta}}(\mathbf{Z}^*)$, the per-node generation log probabilities ($\log p_{\boldsymbol{\theta}}(z_2 \mid \cdot) \ldots \log p_{\boldsymbol{\theta}}(z_M \mid \cdot)$), and also gradients of LM adaptors: $\nabla_{\boldsymbol{\theta}_k} \log \tilde{p}_{\boldsymbol{\theta}}(\mathbf{Z}^*)$ for adaptor hyperedges' indices $k$. We note that `backward` is easily parallelizable: all adaptor edges can be processed at the same time.

---

**Algorithm 2** TAC Backward Algorithm (`backward`)

---

**Input:** $C = (\mathbf{Z}, \mathbf{E})$ and sample $\mathbf{Z}^* = \{\mathbf{z}_1^*, \mathbf{z}_2^*, \ldots, \mathbf{z}_M^*\}$
**Output:** $(\log \tilde{p}_{\boldsymbol{\theta}}(\mathbf{Z}^*), \ (\log p_{\boldsymbol{\theta}}(z_2 \mid \cdot) \ldots \log p_{\boldsymbol{\theta}}(z_M \mid \cdot)), \ \{\nabla_{\boldsymbol{\theta}_k} \log \tilde{p}_{\boldsymbol{\theta}}(\mathbf{Z}^*) \mid e_k \in$
$\mathbf{E}$ is an adaptor hyperedge$\})$
  1: Initialize log-probability accumulator $\mathcal{L} \leftarrow 0$.
  2: **for** each LM adaptor hyperedge $e_k = (\tau_i, \tau_o, \boldsymbol{\theta}_k)$ **do**
  3:    Let $\mathbf{z}_i^* \in \mathbf{Z}^*$, $\mathbf{z}_o^* \in \mathbf{Z}^*$ be the sample value of $e_k$'s input and output nodes $\mathbf{z}_i$ (typed $\tau_i$) and
     $\mathbf{z}_o$ respectively.
  4:    $(\ell, \mathbf{g}_k) \leftarrow \texttt{peft\_backward}(\log p_{LM}(\mathbf{z}_o^* \mid \texttt{canon}(\texttt{parse}(\mathbf{z}_i^*, \tau_i)); \boldsymbol{\theta})$.
  5:    $\mathcal{L} \leftarrow \mathcal{L} + \ell$
  6:    keep track of $\ell$ by its node index.
  7: **end for**
  8: **# For nodes from deterministic hyperedges, set log prob to $0$ as they have no learnable**
     **parameters.**
  9: **return** $(\mathcal{L}, (\log p_{\boldsymbol{\theta}}(z_2 \mid \cdot) \ldots \log p_{\boldsymbol{\theta}}(z_M \mid \cdot)), \{\mathbf{g}_k \mid e_k \in \mathbf{E}$ is an adaptor hyperedge$\})$.

---

## C.2 TACSTAR

The TACSTaR algorithm (Algorithm 3) takes as input $(C, \{(x_i^*, y_i^*) \mid i \in [1..\mathcal{D}_{\text{train}}]\})$, where $C$ is the TAC to train, and $\{(x_i^*, y_i^*) \mid i \in [1..\mathcal{D}_{\text{train}}]\}$ is the training dataset. As we described in §3.2, TACSTaR uses a 'fallback TAC' heuristics in hope to obtain a sample when the forward algorithm fails.

**Building Fallback TAC.** Given a TAC $C = (\mathbf{Z}, \mathbf{E})$ with input node and output node typed $\tau_i$ and $\tau_o$ respectively, we build its fallback TAC $C_{\text{fallback}} = (\mathbf{Z}', \mathbf{E}')$ (denoted as the function `build_fallback` in Algorithm 3) as follows:

- The input node of $C_{\text{fallback}}$: $\mathbf{z}_1'$ is of the product type $\tau_{io} = \tau_i \times \tau_o$, representing a data container that holds one object of type $\tau_i$ and another object of type $\tau_o$.

- All other nodes $\in \mathbf{Z}$ have their counterpart nodes in $Z'$ (with the same types and indices).

- We copy each hyperedge $e \in \mathbf{E}$ over to $\mathbf{E}'$, connecting nodes with the same indices. In the case that $e$ is a deterministic algorithm hyperedge, and has $\mathbf{z}_1$ as one of its source nodes, we modify the counterpart hyperedge $e'$ to have a deterministic algorithm that first extracts the original object $\texttt{parse}(\mathbf{z}_1)$ from $\texttt{parse}(\mathbf{z}_1')$, and then pass $\texttt{parse}(\mathbf{z}_1)$ to the original algorithm as input.

Adaptors in $C_{\text{fallback}}$ use no-op weights, falling back to the behavior of the base model. We denote such no-op weights as $\boldsymbol{\theta}_0$. For example, Fig. 7 is the $C_{\text{fallback}}$ for Fig. 5b.

## C.3 AMORTIZED TACSTAR

The Amortized TACSTaR algorithm (Algorithm 4) builds upon Algorithm 3 to introduce an inference network TAC. While $C_{\text{fallback}}$ used fixed no-op weights that behave identical to the base language model, Amortized TACSTaR leverages an inference network TAC $C'$ with trainable parameters.

**Building the inference network $C'$.** Given a TAC $C = (\mathbf{Z}, \mathbf{E})$ with input node and output node typed $\tau_i$ and $\tau_o$ respectively, we build the adaptive fallback TAC $C' = (\mathbf{Z}', \mathbf{E}')$ (denoted as the function `build_infer_net` in Algorithm 4). At a high level, every adaptor hyperedge that generates latent variables in $C$ is mapped into a counterpart in $C'$ that also depends on an observed $\tau_i$-typed input and a $\tau_o$-typed output, now encoded as $\mathbf{z}_1'$, typed $\tau_{io}$. Specifically we build $C'$ with the following procedure:

- The input node of $C'$: $\mathbf{z}_1'$ is of the product type $\tau_{io} = \tau_i \times \tau_o$, as with `build_fallback`.

- All nodes $\in \mathbf{Z}$ have their counterpart nodes in $Z'$ (with the same types and indices), except for $\{\mathbf{z}_1, \mathbf{z}_2\}$.[16]

---

[16]We arbitrarily designate a node $\in \mathbf{Z}'$ that does not have an outgoing hyperedge as the output node for syntactic conformity.

---

**Algorithm 3** TACSTaR Training Algorithm

---

**Input:** Training pairs $\mathcal{D}_{\text{train}} = \{(x_i^*, y_i^*) \mid i \in [1..|\mathcal{D}_{\text{train}}|]\}$, TAC $C$, sampler configuration $\kappa$.
1: $C_{\text{fallback}} \leftarrow$ build_fallback($C$)
2: **for** epoch in $[1..\texttt{num\_epochs}]$ **do**
3:      $S \leftarrow \{\}$ # **Successful samples**
4:      **for** training pair $(x^*, y^*) \in \mathcal{D}_{\text{train}}$ **do**
5:          $\mathbf{z}_1^* \leftarrow$ canon($x^*$)
6:          # **E-step (Sampling Latent Variables):**
7:          $(\hat{\mathbf{z}}_2 \ldots \hat{\mathbf{z}}_M) \leftarrow$ Forward$(C, \mathbf{z}_1^*)$.
8:          # **Filtering (Validity Check):**
9:          Initialize error_flag $\leftarrow$ false.
10:         Set error_flag $\leftarrow$ true if errors in E-step or parse($\hat{\mathbf{z}}_2$) $\neq y^*$.
11:         # **Heuristics Fallback (Addressing Forward Failure):**
12:         **if** error_flag is true **then**
13:            $\mathbf{z}_1'^* \leftarrow$ canon$((x^*, y^*))$
14:            $(\hat{\mathbf{z}}_2' \ldots \hat{\mathbf{z}}_M') \leftarrow$ forward$(C_{\text{fallback}}, \mathbf{z}_1'^*)[0]$.
15:            **if** no error was raised and parse($\hat{\mathbf{z}}_2'$) $= y^*$ **then**
16:               $(\hat{\mathbf{z}}_2 \ldots \hat{\mathbf{z}}_M) \leftarrow (\hat{\mathbf{z}}_2' \ldots \hat{\mathbf{z}}_M')$
17:               Set error_flag $\leftarrow$ false.
18:            **end if**
19:         **end if**
20:         **if** error_flag is false **then**
21:            $S \leftarrow S \cup \{(\mathbf{z}_1^*, \hat{\mathbf{z}}_2 \ldots \hat{\mathbf{z}}_M)\}$
22:         **end if**
23:      **end for**
24:      # **M-step (Parameter Update):**
25:      **for** $(\mathbf{z}_1^*, \hat{\mathbf{z}}_2 \ldots \hat{\mathbf{z}}_M) \in S$ **do**
26:          $\mathbf{G} \leftarrow$ backward$(C, (\mathbf{z}_1^*, \hat{\mathbf{z}}_2 \ldots \hat{\mathbf{z}}_M))[2]$
27:          optimize$(C, \mathbf{G})$
28:      **end for**
29: **end for**

---

- For each hyperedge $e \in \mathbf{E}$,
  - In the case that $e$ is a deterministic algorithm hyperedge, and has $\mathbf{z}_1$ as one of its source nodes, we add a counterpart hyperedge $e'$ that connect counterpart nodes in $\mathbf{Z}'$, with its deterministic algorithm modified to typecheck, as build_fallback.
  - Otherwise, $e$ is an adaptor hyperedge. Denoting its source node as $\mathbf{z}_s$ and target node as $\mathbf{z}_t$:
    * If $\mathbf{z}_t = \mathbf{z}_2$, we continue since $\mathbf{z}_t$ has no counterpart $C'$.
    * If $\mathbf{z}_s = \mathbf{z}_1$ and $\mathbf{z}_t \neq \mathbf{z}_2$, we add a counterpart hyperedge $e' = (\tau_s, \tau_t, \boldsymbol{\theta}_{\text{new}})$ connecting counterpart nodes $\mathbf{z}_s'$ and $\mathbf{z}_t'$. $\boldsymbol{\theta}_{\text{new}}$ indicates the parameter vector of a new LM adaptor.
    * Otherwise, $\mathbf{z}_s \neq \mathbf{z}_1$ and $\mathbf{z}_t \neq \mathbf{z}_2$. In this case, we create $e'$ to be an adaptor that is conditioned on both $\mathbf{z}_s'$ and $\mathbf{z}_1'$. To achieve this goal, we introduce into $C'$ a helper node $\mathbf{z}_s''$ typed $\tau_{ios} = \tau_i \times \tau_o \times \tau_s$, and a helper hyperedge $e''$ that has source nodes $\{\mathbf{z}_1', \mathbf{z}_s'\}$, and target node $\{\mathbf{z}_s''\}$. $e''$ is a deterministic edge that combines values in $\mathbf{z}_1'$ and $\mathbf{z}_s'$ into the 3-object container $\mathbf{z}_s''$. Finally, we add $e'$ that connects $\mathbf{z}_s''$ to $\mathbf{t}$ as the adaptor transformation $(\tau_{ios}, \tau_t, \boldsymbol{\theta}_{\text{new}})$, where $\boldsymbol{\theta}_{\text{new}}$ again indicates the parameter vector of a new LM adaptor.

Adaptors in $C'$ are new adaptors. And we train $C$ alternately with $C'$ in Algorithm 4. The algorithm to train $C'$ is listed in Algorithm 5.

## C.4 UPDATING $C'$

We train the inference network $C'$ to better approximate the posterior distribution defined by $C$ alternately (§3.3). In other words, we update adaptor parameters in $C'$ so that sampled latent

---

**Algorithm 4** Amortized TACSTaR Training Algorithm

---

**Input:** Training pairs $\mathcal{D}_{\text{train}} = \{(x_i^*, y_i^*) \mid i \in [1..|\mathcal{D}_{\text{train}}|]\}$, TAC $C$, sampler configuration $\kappa$.
1: $C' \leftarrow \texttt{build\_infer\_net}(C)$
2: **for** epoch in $[1..\texttt{num\_epochs}]$ **do**
3:      $S \leftarrow \{\}$ # **Successful samples**
4:      **for** training pair $(x^*, y^*) \in \mathcal{D}_{\text{train}}$ **do**
5:          $\mathbf{z}_1^* \leftarrow \texttt{canon}(x^*)$
6:          # **E-step (Sampling Latent Variables):**
7:          $(\hat{\mathbf{z}}_2 \ldots \hat{\mathbf{z}}_M) \leftarrow \text{Forward}(C, \mathbf{z}_1^*)$.
8:          # **Filtering (Validity Check):**
9:          Initialize $\texttt{error\_flag} \leftarrow \text{false}$.
10:         Set $\texttt{error\_flag} \leftarrow \text{true}$ if errors in E-step or $\texttt{parse}(\hat{\mathbf{z}}_2) \neq y^*$.
11:         # **Heuristics Fallback (Addressing Forward Failure):**
12:         **if** $\texttt{error\_flag}$ is true **then**
13:            $\mathbf{z}_1'^* \leftarrow \texttt{canon}((x^*, y^*))$
14:            $(\hat{\mathbf{z}}_2' \ldots \hat{\mathbf{z}}_M') \leftarrow \texttt{forward}(C_{\text{fallback}}, \mathbf{z}_1'^*)[0]$.
15:            **if** no error was raised and $\texttt{parse}(\hat{\mathbf{z}}_2') = y^*$ **then**
16:               $(\hat{\mathbf{z}}_2 \ldots \hat{\mathbf{z}}_M) \leftarrow (\hat{\mathbf{z}}_2' \ldots \hat{\mathbf{z}}_M')$
17:               Set $\texttt{error\_flag} \leftarrow \text{false}$.
18:            **end if**
19:         **end if**
20:         **if** $\texttt{error\_flag}$ is true **then**
21:            $(\hat{\mathbf{z}}_3 \ldots \hat{\mathbf{z}}_M) \leftarrow \texttt{forward}(C', \mathbf{z}_1'^*)[0]$
22:            Set $\texttt{error\_flag} \leftarrow \text{false}$ if no errors in previous step.
23:         **end if**
24:         **if** $\texttt{error\_flag}$ is false **then**
25:            $S \leftarrow S \cup \{(\mathbf{z}_1^*, \mathbf{z}_2^*, \hat{\mathbf{z}}_3, \ldots \hat{\mathbf{z}}_M)\}$
26:         **end if**
27:      **end for**
28:      # **M-step (Parameter Update):**
29:      **for** $(\mathbf{z}_1^*, \hat{\mathbf{z}}_2 \ldots \hat{\mathbf{z}}_M) \in S$ **do**
30:          $\mathbf{G} \leftarrow \texttt{backward}(C, (\mathbf{z}_1^*, \hat{\mathbf{z}}_2 \ldots \hat{\mathbf{z}}_M))[2]$
31:          $\texttt{optimize}(C, \mathbf{G})$
32:      **end for**
33:      $C' \leftarrow$ update inference network $C'$ (§C.4).
34: **end for**

---

variables of $C'$ $((\hat{\mathbf{z}}_3, \ldots, \hat{\mathbf{z}}_M)$ obtained using $\texttt{forward}(C', \texttt{canon}(x^*), \kappa))$ follow the normalized distributions under $C$ (obtained using $\texttt{backward}(C, (\texttt{canon}(x^*), \texttt{canon}(y^*), \hat{\mathbf{z}}_3, \ldots, \hat{\mathbf{z}}_M)))$. To promote diversity of samples, we additionally obtain samples from $C_{\text{fallback}}$ (§C.2). Let $\mathbf{Z} = (\mathbf{z}_3^*, \ldots, \mathbf{z}_M^*)$ be a sample out of $G$ collected samples $(\mathbf{Z}^{(1)}, \ldots, \mathbf{Z}^{(G)})$ from $C_{\text{fallback}}$ and $C'$. We approximate the posterior probability of $\mathbf{Z}$ under $C$, conditioning on $\mathbf{z}_1^* = \texttt{canon}(x^*)$, $\mathbf{z}_2^* = \texttt{canon}(y^*)$ under the balance heuristic (Veach & Guibas, 1995) as

$$\hat{p}_{\text{posterior}}(\mathbf{Z}) \propto \frac{(N_{\text{fallback}} + N_{\text{infer}})\tilde{p}_{\text{model}}}{N_{\text{fallback}} p_{\text{fallback}} + N_{\text{infer}} p_{\text{infer}}}, \quad (5)$$

where $\tilde{p}_{\text{model}} = \tilde{p}_C(\mathbf{z}_1^*, \mathbf{z}_2^*, \mathbf{z}_3^*, \ldots, \mathbf{z}_M^*)$, $p_{\text{fallback}} = \prod_{m=3}^{M} p_{LM}(\mathbf{z}_m^* \mid \mathbf{z}_m^*\text{'s source node}; \boldsymbol{\theta}_0)$, and $p_{\text{infer}} = \prod_{m=3}^{M} p_{LM}(\mathbf{z}_m^* \mid \mathbf{z}_m^*\text{'s source node}; \boldsymbol{\theta}_{\text{new}})$. These values are all obtained using the $\texttt{backward}$ algorithm.[17] We denote the number of samples attempted (including errors) on $C_{\text{fallback}} = N_{\text{fallback}}$, the number of samples attempted (including errors) on $C' = N_{\text{infer}}$. $\hat{p}_{\text{posterior}}$ is normalized over the mixture so that $\sum_{g=1}^{G} \hat{p}_{\text{posterior}}(\mathbf{Z}^{(g)}) = 1$.

---

[17] $\texttt{backward}$ algorithm as presented in this work computes both gradients and probabilities. In our implementation we do not compute gradients when they are not needed; but we omit this subtlety in Algorithm 2.

Algorithm 5 updates adaptors in $C'$ to bring its unnormalized distribution closer to Eq. (5). Since the self-normalized approximation of the posterior distribution is consistent but biased, we require minimum numbers of samples from $C'$ and $C_{\text{fallback}}$.

---

**Algorithm 5** `update_infer_net`

---

**Input:** Training pair $(x^*, y^*)$, model TAC $C$, sampler configuration $\kappa$, inference network $C'$, non-adaptive fallback $C_{\text{fallback}}$, number of samples from $C_{\text{fallback}}$: $G_{\text{fallback}}$, number of samples from $C'$: $G_{\text{infer}}$ .

1: $\mathbf{z}_1'^* \leftarrow \texttt{canon}((x^*, y^*))$, $\mathbf{z}_1^* \leftarrow \texttt{canon}(x^*)$, $\mathbf{z}_2^* \leftarrow \texttt{canon}(y^*)$.
2: $\mathbf{Z}_{\text{collected}} \leftarrow []$
3: **# In our implementation we give up and raise an error after** 30 **unsuccessful attempts.**
4: **while** number of successful samples from $C_{\text{fallback}} < G_{\text{fallback}}$ **do**
5:     Try $(\hat{\mathbf{z}}_2, \hat{\mathbf{z}}_3, \ldots \hat{\mathbf{z}}_M) \leftarrow \texttt{forward}(C_{\text{fallback}}, \mathbf{z}_1'^*, \kappa, 1)$
6:     **if** previous step succeeded **then**
7:         **# We discard** $\hat{\mathbf{z}}_2$ **from** $C_{\text{fallback}}$.
8:         Append $(\hat{\mathbf{z}}_3, \ldots, \hat{\mathbf{z}}_M)$ to $\mathbf{Z}_{\text{collected}}$.
9:     **end if**
10: **end while**
11: $N_{\text{fallback}} \leftarrow$ numbers of attempts on $C_{\text{fallback}}$
12: **while** number of successful samples from $C' < G_{\text{infer}}$ **do**
13:     Try $(\hat{\mathbf{z}}_3, \ldots \hat{\mathbf{z}}_M) \leftarrow \texttt{forward}(C', \mathbf{z}_1'^*, \kappa, 1)$
14:     **if** previous step succeeded **then**
15:         Append $(\hat{\mathbf{z}}_3, \ldots, \hat{\mathbf{z}}_M)$ to $\mathbf{Z}_{\text{collected}}$.
16:     **end if**
17: **end while**
18: $N_{\text{infer}} \leftarrow$ numbers of attempts on $C'$
19: $G \leftarrow G_{\text{fallback}} + G_{\text{infer}}$
20: Assert $G = |\mathbf{Z}_{\text{collected}}|$
21: Compute $[\hat{p}_{\text{posterior}}(\mathbf{Z}^{(1)}) \ldots \hat{p}_{\text{posterior}}(\mathbf{Z}^{(G)})]$ using Eq. (5).
22: Sample $g \in [1..G]$ with probability proportional to $\hat{p}_{\text{posterior}}(\mathbf{Z}^{(g)}$.
23: $\mathbf{G} \leftarrow \texttt{backward}(C', \mathbf{Z}^{(g)})[2]$.
24: $\texttt{optimize}(C', \mathbf{G})$

---

## D   FORMAL STATEMENTS AND PROOFS REGARDING TYPE COMPLIANCE

**Well-specifiedness.** Let $C = (\mathbf{Z}, \mathbf{E})$. We define well-specifiedness for TAC: we say $\boldsymbol{\theta} = \{\boldsymbol{\theta}_1 \ldots \boldsymbol{\theta}_K\}$ is well-specified if for every LM adaptor $e_k = (\tau_i, \tau_o, \boldsymbol{\theta}_k) \in \mathbf{E}$ and for every valid canonicalized string $\boldsymbol{x}$ of type $\tau_i$, the LM distribution $p_{LM}$ only has support over valid outputs of type $\tau_o$. Formally, $\forall$ valid $x$, $\sum_{\boldsymbol{y} \in \mathcal{D}_{\text{valid}}(\tau_o)} p_{LM}(\boldsymbol{y} \mid \boldsymbol{x}; \boldsymbol{\theta}_k) = 1$ iff $\boldsymbol{\theta}$ is well-specified.

We first prove that hyperedges are locally normalized (*i.e.*, the partition function is 1) when $\boldsymbol{\theta}$ is well-specified:

**Lemma 1.** *If $\boldsymbol{\theta}$ is well-specified, then for any hyperedge $e_k \in \mathbf{E}$ and any valid assignment $\boldsymbol{x}$ to its source nodes, the local partition function $Z_k = 1$.*

*Proof.* $e_k$ is either an LM adaptor or a deterministic algorithm:

- If $e_k$ is an LM adaptor, $Z_k = \sum_{\boldsymbol{y}} \tilde{p}_{\boldsymbol{\theta}}(\boldsymbol{y} \mid \boldsymbol{x}; e_k) = \sum_{\boldsymbol{y} \in \text{valid}(\tau_o)} p_{LM}(\boldsymbol{y} \mid \boldsymbol{x}; \boldsymbol{\theta}_k) = 1$.

- If $e_k$ is a deterministic algorithm, by Eq. (2) $Z_k = \sum_{\boldsymbol{y}} \tilde{p}(\boldsymbol{y} \mid \boldsymbol{x}; e_k) = \tilde{p}(\texttt{canon}(f(\texttt{parse}(\boldsymbol{x}, \tau_i)))) + 0 = 1 + 0 = 1$.

$\square$

We then use induction based on the TAC $C$'s topological structure.

**Lemma 2.** *Let $\boldsymbol{\theta}$ be a well-specified parameter vector for* TAC $C = (\mathbf{Z}, \mathbf{E})$. *The conditional partition function $\mathcal{Z}_{\boldsymbol{\theta}}(\mathbf{z}_1) = 1$.*

*Proof.* We use induction on the number of nodes $k$, following the topological sort $\mathbf{z}_1, \ldots \mathbf{z}_M$. For clarity, here we abuse the subscript notation for topological order, and therefore $\mathbf{z}_M$ (instead of $\mathbf{z}_2$) is the output.

Let $C_k$ be the sub-TAC induced by $\{\mathbf{z}_1, \ldots, \mathbf{z}_k\}$. Its partition function is $\mathcal{Z}_k(\mathbf{z}_1) = \sum_{\mathbf{z}_2 \ldots \mathbf{z}_k} \prod_{m=2}^{k} \tilde{p}_{\boldsymbol{\theta}}(\mathbf{z}_m \mid S_m)$, where $S_m$ denotes the source nodes of $\mathbf{z}_m$ under its corresponding hyperedge.

**Base Case.** $k = 1$. $C_1$ has only $\mathbf{z}_1$. $\mathcal{Z}_1(\mathbf{z}_1) = 1$ since the product is empty.

**Inductive Step.** We assume $\mathcal{Z}_{k-1}(\mathbf{z}_1) = 1$. First we rewrite $\mathcal{Z}_k(\mathbf{z}_1)$ by explicitly summing over $\mathbf{z}_k$. Since $\mathbf{z}_1, \ldots \mathbf{z}_k$ is a topological order, the source nodes of $\mathbf{z}_k$: $S_k$ is a subset of $\{\mathbf{z}_1, \ldots \mathbf{z}_{k-1}\}$. We thus rewrite $\mathcal{Z}_k(\mathbf{z}_1)$ as

$$\mathcal{Z}_k(\mathbf{z}_1) = \sum_{\mathbf{z}_2 \ldots \mathbf{z}_k} \left( \prod_{m=2}^{k-1} \tilde{p}_{\boldsymbol{\theta}}(\mathbf{z}_m \mid S_m) \right) \cdot \left( \sum_{\mathbf{z}_k} \tilde{p}_{\boldsymbol{\theta}}(\mathbf{z}_k \mid S_k) \right). \tag{6}$$

We discuss the summands by the validity of $\mathbf{z}_2 \ldots \mathbf{z}_{k-1}$:

- If $\mathbf{z}_2 \ldots \mathbf{z}_{k-1}$ is valid: by Lemma 1 the term $\sum_{\mathbf{z}_k} \tilde{p}_{\boldsymbol{\theta}}(\mathbf{z}_k \mid S_k) = 1$. This summand is therefore $\prod_{m=2}^{k-1} \tilde{p}_{\boldsymbol{\theta}}(\mathbf{z}_m \mid S_m)$.

- If $\mathbf{z}_2 \ldots \mathbf{z}_{k-1}$ is not valid: by Eqs 1 and 2 this summand is 0.

We can thus rewrite Eq. (6) as

$$\mathcal{Z}_k(\mathbf{z}_1) = \sum_{\mathbf{z}_2, \ldots, \mathbf{z}_{k-1} \mid \text{valid assignments}} \prod_{m=2}^{k-1} \tilde{p}_{\boldsymbol{\theta}}(\mathbf{z}_m \mid S_m). \tag{7}$$

Equation (7) can be further rewritten to sum over both valid and invalid $\mathbf{z}_2, \ldots, \mathbf{z}_{k-1}$ assignments (since again by Eqs. (1) and (2), the summand is 0 for invalid assignments):

$$\mathcal{Z}_k(\mathbf{z}_1) = \sum_{\mathbf{z}_2, \ldots, \mathbf{z}_{k-1}} \prod_{m=2}^{k-1} \tilde{p}_{\boldsymbol{\theta}}(\mathbf{z}_m \mid S_m) = \mathcal{Z}_{k-1}(\mathbf{z}_1). \tag{8}$$

Since by assumption $\mathcal{Z}_{k-1}(\mathbf{z}_1) = 1$, we thus prove by induction $\mathcal{Z}_M(\mathbf{z}_1) = \mathcal{Z}_{\boldsymbol{\theta}}(\mathbf{z}_1) = 1$. $\qquad\square$

Finally, we show that Lemma 2 implies the equivalence of maximizing the normalized and unnormalized likelihoods when the true parameters are well-specified.

**Theorem 1.** *Let $\boldsymbol{\Theta}$ be the entire parameter space and let $\boldsymbol{\Theta}' \subseteq \boldsymbol{\Theta}$ be the subset of well-specified parameters. Assume $\boldsymbol{\theta}^*$ uniquely maximizes the normalized likelihood $p_{\boldsymbol{\theta}}(\mathbf{z}_{2..M} | \mathbf{z}_1)$ and resides $\in \boldsymbol{\Theta}'$. Then, $\hat{\boldsymbol{\theta}} = \arg\max_{\boldsymbol{\theta} \in \boldsymbol{\Theta}} \tilde{p}_{\boldsymbol{\theta}}(\mathbf{z}_{2..M} | \mathbf{z}_1) \implies \hat{\boldsymbol{\theta}} = \boldsymbol{\theta}^*$.*

*Proof.* First we note $\forall \boldsymbol{\theta} \in \boldsymbol{\Theta}, \mathcal{Z}_{\boldsymbol{\theta}}(\mathbf{z}_1) \leq 1$, since for any adaptor $\sum_y \tilde{p}_{\boldsymbol{\theta}}(\mathbf{y} \mid \mathbf{x}) \leq 1$. By Eqs. (1) and (2) the global partition function must also be $\leq 1$.

We rewrite the unnormalized likelihood as a product of normalized likelihood and the partition function:

$$\tilde{p}_{\boldsymbol{\theta}}(\mathbf{z}_{2 \ldots M} \mid \mathbf{z}_1) = p_{\boldsymbol{\theta}}(\mathbf{z}_{2 \ldots M} \mid \mathbf{z}_1) \cdot \mathcal{Z}_{\boldsymbol{\theta}}(\mathbf{z}_1) \tag{9}$$

Since $\mathcal{Z}_{\boldsymbol{\theta}}(\mathbf{z}_1) \leq 1, \forall \boldsymbol{\theta} \in \boldsymbol{\Theta}, \tilde{p}_{\boldsymbol{\theta}}(\mathbf{z}_{2 \ldots M} \mid \mathbf{z}_1) \leq p_{\boldsymbol{\theta}}(\mathbf{z}_{2 \ldots M} \mid \mathbf{z}_1)$.

At the well-specified true parameters $\boldsymbol{\theta}^*$ we have $\mathcal{Z}_{\boldsymbol{\theta}}(\mathbf{z}_1) = 1$ by Lemma 2. Therefore $\tilde{p}_{\boldsymbol{\theta}^*}(\mathbf{z}_{2\ldots M} \mid \mathbf{z}_1) = p_{\boldsymbol{\theta}^*}(\mathbf{z}_{2\ldots M} \mid \mathbf{z}_1)$.

By our assumption that $\boldsymbol{\theta}^*$ maximizes normalized likelihood, $\forall \boldsymbol{\theta} \in \boldsymbol{\Theta}, p_{\boldsymbol{\theta}^*}(\mathbf{z}_{2\ldots M} \mid \mathbf{z}_1) \geq p_{\boldsymbol{\theta}}(\mathbf{z}_{2\ldots M} \mid \mathbf{z}_1)$.

Combining everything together:

$$\begin{aligned}
\tilde{p}_{\boldsymbol{\theta}^*}(\mathbf{z}_{2\ldots M} \mid \mathbf{z}_1) &= p_{\boldsymbol{\theta}^*}(\mathbf{z}_{2\ldots M} \mid \mathbf{z}_1) \\
&\geq p_{\boldsymbol{\theta}}(\mathbf{z}_{2\ldots M} \mid \mathbf{z}_1) \\
&\geq \tilde{p}_{\boldsymbol{\theta}}(\mathbf{z}_{2\ldots M} \mid \mathbf{z}_1)
\end{aligned}$$

for all $\boldsymbol{\theta} \in \boldsymbol{\Theta}$. Under the assumption $\boldsymbol{\theta}^*$ is unique, $\boldsymbol{\theta}^* = \arg\max_{\boldsymbol{\theta} \in \boldsymbol{\Theta}} \tilde{p}_{\boldsymbol{\theta}}(\mathbf{z}_{2\ldots M} \mid \mathbf{z}_1) = \hat{\boldsymbol{\theta}}$. $\qquad\square$

**Theorem 2.** *Let $\boldsymbol{\theta} = \{\boldsymbol{\theta}_1 \ldots \boldsymbol{\theta}_K\}$ be the union of a $K$-adaptor* TAC*'s LM adaptor parameters . If $\forall \mathbf{z}_{k,1} \in \Sigma^*, \mathbf{z}_{k,2} \in \Sigma^*, \|\nabla\boldsymbol{\theta}\left(\sum \log p_{LM}(\mathbf{z}_{k,2} \mid \mathbf{z}_{k,1}; \boldsymbol{\theta})\right)\|_\infty \leq G$, then $\nabla_{\boldsymbol{\theta}} \log \mathcal{Z}_{\boldsymbol{\theta}} \leq 2G(1 - \mathcal{Z}_{\boldsymbol{\theta}})$.*

*Proof.* Here we fix $\mathbf{z}_1 = x$. We denote $\mathbf{z}_{2\ldots M} = y$. Let $p_{LM}^{(k)}(y)$ be the $k$-th LM adaptor's unmasked node probability, given $(x, y)$ as TAC input and output. We then denote $p_{\boldsymbol{\theta}}(y) = \prod_k p_{LM}^{(k)}$ as a TAC's *normalized* distribution over node assignments (without masking invalid ones). The partition function $\mathcal{Z}_{\boldsymbol{\theta}} = \sum_y p_{\boldsymbol{\theta}}(y \mid x)\mathbb{I}(y \in V) = \Pr_{p_{\boldsymbol{\theta}}}(V)$ where $V$ is the set of valid node assignments.

We first rewrite $\nabla_{\boldsymbol{\theta}} \log \mathcal{Z}_{\boldsymbol{\theta}}$ as an expectation under $p_{\boldsymbol{\theta}}$:

$$\nabla_{\boldsymbol{\theta}} \log \mathcal{Z}_{\boldsymbol{\theta}} = \mathbb{E}_{y \sim p_{\boldsymbol{\theta}}(\cdot \mid V)}\left[\nabla_{\boldsymbol{\theta}} \log p_{\boldsymbol{\theta}}(y)\right]. \tag{10}$$

Using the identity $\sum_y p_{\boldsymbol{\theta}}(y)\nabla_{\boldsymbol{\theta}} \log p_{\boldsymbol{\theta}}(y) = 0$, we rewrite Eq. (10) as

$$\nabla_{\boldsymbol{\theta}} \log \mathcal{Z}_{\boldsymbol{\theta}} = \mathbb{E}_{y \sim p_{\boldsymbol{\theta}}(\cdot \mid V)}\left[\nabla_{\boldsymbol{\theta}} \log p_{\boldsymbol{\theta}}(y)\right] - \mathbb{E}_{y \sim p_{\boldsymbol{\theta}}}\left[\nabla_{\boldsymbol{\theta}} \log p_{\boldsymbol{\theta}}(y)\right]. \tag{11}$$

Let $f = \nabla_{\boldsymbol{\theta}} \log p_{\boldsymbol{\theta}}(y)$. We can now rewrite $\|\nabla_{\boldsymbol{\theta}} \log \mathcal{Z}_{\boldsymbol{\theta}}\|_\infty$ as

$$\begin{aligned}
\|\nabla_{\boldsymbol{\theta}} \log \mathcal{Z}_{\boldsymbol{\theta}}\|_\infty &= \|\mathbb{E}_{p_{\cdot \mid V}}\left[f\right] - \mathbb{E}_{p_{\boldsymbol{\theta}}}\left[f\right]\|_\infty \\
&= \|\sum_y f \cdot (p_{\boldsymbol{\theta}}(y \mid V) - p_{\boldsymbol{\theta}}(y))\|_\infty \\
&\leq \sum_y \|f\|_\infty \cdot |p_{\boldsymbol{\theta}}(y \mid V) - p_{\boldsymbol{\theta}}(y)| \\
&\leq \sum_y G \cdot |p_{\boldsymbol{\theta}}(y \mid V) - p_{\boldsymbol{\theta}}(y)|. \tag{12}
\end{aligned}$$

Noting that $\sum_y |p_{\boldsymbol{\theta}}(y \mid V) - p_{\boldsymbol{\theta}}(y)|$ is twice the total variation between $p_{\boldsymbol{\theta}}$ and $p_{\boldsymbol{\theta}}(\cdot \mid V)$, and that the total variation between $p_{\boldsymbol{\theta}}$ and $p_{\boldsymbol{\theta}}(\cdot \mid V)$ is $(1 - \mathcal{Z}_{\boldsymbol{\theta}})$ — the sum of invalid assignments' probabilities under $p_{\boldsymbol{\theta}}$ — we can rewrite Eq. (12) as $\|\nabla_{\boldsymbol{\theta}} \log \mathcal{Z}_{\boldsymbol{\theta}}\|_\infty \leq 2G(1 - \mathcal{Z}_{\boldsymbol{\theta}})$.

$\square$

# E  IMPLEMENTATION CONSIDERATIONS

In this section we discuss practical implementation considerations. In particular, we distinguish between *one-time* and *per-use* efforts.

## E.1  ONE-TIME EFFORTS

**Parsing and canonicalization.** There exist multiple libraries that can readily be used to implement `parse` and `canon` for typed data-holding objects in Python. One example is Langfun which we use extensively in the paper. Another popular library is Pydantic, which is used in DSPy.

**Type validation logic.** As we briefly discussed in Footnote 5, the `parse` function can be used to implement complex business logic. Such logic can usually be implemented cleanly as part of type definition (*e.g.*, as `__init__` and `__post_init__` methods in Python).

**Algorithms.**    The core TAC algorithms for execution and training (Algorithms listed in §C) are general and need only be implemented once. The main computational bottlenecks in these algorithms are:

- Sampling from an LM adaptor $p_{LM}(\cdot; \boldsymbol{\theta})$.
- Evaluating the conditional probability of $\boldsymbol{y}$ given $\boldsymbol{x}$ under an LM adaptor: $p_{LM}(\boldsymbol{y} \mid \boldsymbol{x}; \boldsymbol{\theta})$.
- Computing gradients of $(\boldsymbol{x}, \boldsymbol{y})$ with regard to parameters $\boldsymbol{\theta}$: $\nabla_{\boldsymbol{\theta}} \log p_{LM}(\boldsymbol{y} \mid \boldsymbol{x}; \boldsymbol{\theta})$.

A practical implementation can abstract these bottlenecks away, by offloading these intensive parts to dedicated inference servers (e.g., vLLM). The core TAC logic remains a lightweight, accelerator-agnostic program. Furthermore, since TACs use parameter-efficient fine-tuning (PEFT), the adaptor weights and gradients are small enough to be processed quickly, often without needing dedicated accelerators for the logic itself. This design significantly reduces the low-level engineering burden.

### E.2 PER-USE EFFORTS

Once the core engine is in place, a practitioner's effort is focused on defining a TAC hypergraph for their specific task. Since the TAC hypergraph is essentially a data flow graph, it can be represented in a way that is directly analogous to network architecture definitions in popular neural network frameworks such as PyTorch, where the `Module` s represent hyperedges, and their `forward` methods connect the typed data nodes.

## F    ADDITIONAL TAC DIAGRAMS OF TRAINABLE WORKFLOWS

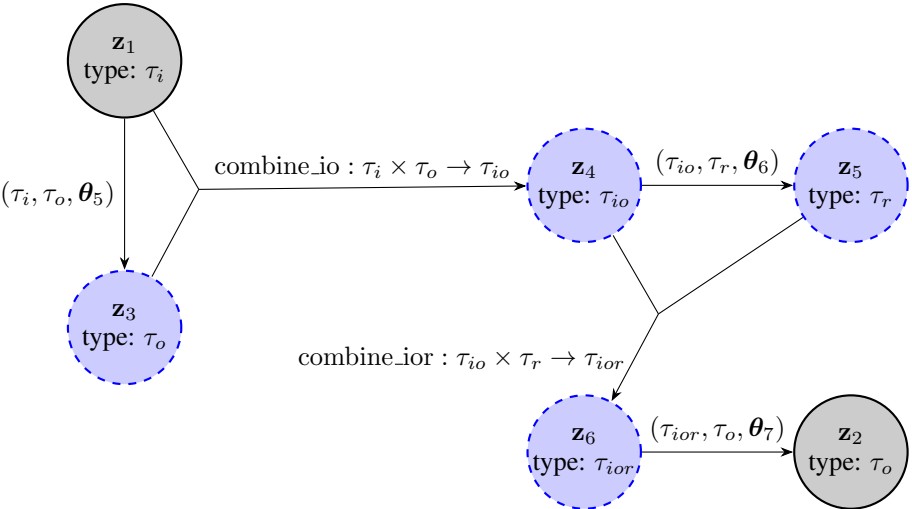

Figure 6: **refine-structure**: refinement through cascade topology engineering. This cascade models a refinement process where an initial output sketch is iteratively refined based on generated rationales.

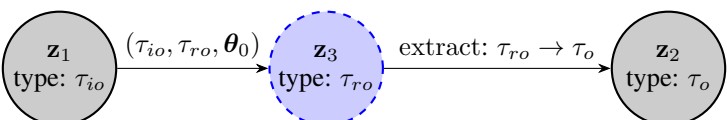

Figure 7: $C_{\text{fallback}}$ for **cot-type-structure**. Notice that the adaptor $(\tau_{io}, \tau_{ro}, \boldsymbol{\theta}_0)$ uses 'fallback' weights $\boldsymbol{\theta}_0$ that represent no-op weights. Since we conduct experiment on LoRA adaptors in this work, we use the zero-init vectors as $\boldsymbol{\theta}_0$.

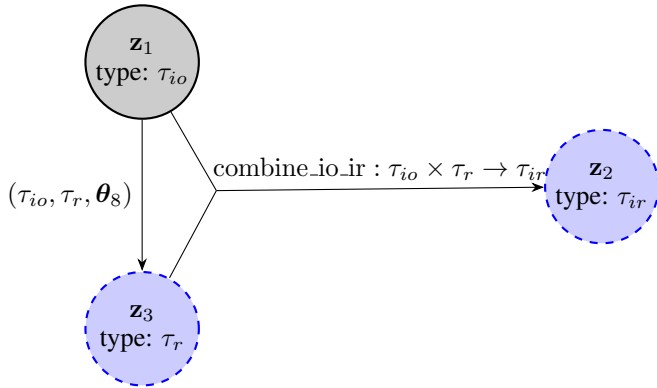

Figure 8: Inference network TAC $C'$ for **cot-type-structure**.

## G   FURTHER DETAILS OF EXPERIMENT SETUP

**Data splits.**   We focus on the low-data regime of task adaptation in this work. For MGSM and MGSM-SymPy, each language has $100/30/120$ training/validation/test examples respectively. The splits are $100/30/100$ and $100/30/300$ for HotPotQA and FinQA respectively. For HotPotQA and FinQA, we use the first entries from the original dataset files as our training and evaluation subsets. For MGSM experiments, we train and evaluate on each language separately. For MuSR tasks, the splits are $100/30/120$ and $100/30/126$ respectively.

**Evaluation.**   We look at exact match accuracy scores of the answers for all 5 tasks. For MGSM-SymPy experiments, we convert answers from the dataset to integers; as for the model predictions, we evaluate the expressions as rational numbers under SymPy[18], and cast the results as integer numbers. We do not make use of additional clues from the datasets (*e.g.*, the rationales provided for the 8 examples in MGSM datasets).

### G.1   TAC SETUP

**Training procedure.**   We train all workflows that have latent variables with our TACSTaR and Amortized TACSTaR algorithms, except for the original (untyped) STaR experiments. Since **direct** experiments do not have latent variables, we train those models using the ordinary cross entropy loss. In all experiments we use a batch size of $8$. The Adam optimizer (Kingma & Ba, 2014) is used throughout all experiments, with a learning rate of $5e-5$. We early-stop if no higher validation score is achieved for $4$ consecutive epochs. The sampler configuration $\kappa$ is set to use a combination of top-K and nucleus sampling (Holtzman et al., 2020), where we first choose the top $40$ candidates, and cut off accumulated probability mass at $0.95$. To train the inference TACs, we accumulate $32$ samples from $C_{\text{infer}}$ and $16$ samples from the fallback model (that is, $G=48$ at the end of Algorithm 5).

**Decoding procedure for generation tasks.**   Here we denote the answer type as $\tau_o$. For each test input instance, we obtain $32$ samples $\hat{\mathbf{Z}}^{(1)}\ldots\hat{\mathbf{Z}}^{(32)}$ using forward, bucket their output node values $\texttt{parse}(\hat{\mathbf{z}}_2^{(1)},\tau_o)\ldots\texttt{parse}(\hat{\mathbf{z}}_2^{(32)},\tau_o)$ into $B$ bins, identified by the parsed output $y_1\ldots y_B$. We output the answer with maximum accumulated unnormalized probability mass, namely $\arg\max_b \sum_{s\in[1..32],\texttt{parse}(\hat{\mathbf{z}}_2,\tau_o)=y_b} \tilde{p}_{\boldsymbol{\theta}}(\hat{\mathbf{Z}}^{(s)})$.

**Decoding procedure for classification tasks.**   We estimate each label $c$'s normalized marginal probability using Eq. (15), with $N=32$. We output the label with largest normalized marginal probability as prediction.

**Object representation of data.**   We represent input $\tau_i$ and output $\tau_o$ as Python types. The objects are encoded as string representations under Langfun. We design the input and output types separately

---

[18]https://www.sympy.org/en/index.html

to reflect the original dataset schemata (Listings 1 to 3). As for the rationales (represented by $\tau_r$ in **cot-type-structure** and **cot-cascade-structure**) we represent them as lists of strings (Listing 4). Product types are represented as new Python classes (*e.g.*, the product of type `Question` and `Answer`, represented as $\tau_{io}$ in Figs. 7 and 8, is a new class QuestionAnswer). The object representation can be arbitrarily complex, with Langfun handling all `canon` and `parse` logic (for example, Listing 6 has `Answer` objects embedded in multiple types; and Listing 7 has self-referential definitions).

```python
class Question:
  question: str

class Answer:
  answer: str
```

Listing 1: Input and output type definitions for MGSM

```python
class Paragraph:
  title: str
  sentences: list[str]

class Context:
  paragraphs: list[Paragraph]

class Answer:
  answer: str

class Question:
  id: str
  question: str
  context: Context
```

Listing 2: Input and output type definitions for HotPotQA

```python
class Question:
  question: str
  pre_text: list[str]
  table: list[list[str]]
  post_text: list[str]

class Step:
  op: str
  arg1: str
  arg2: str
  res: str

class Answer:
  answer: str

class QuestionAnswer:
  question: Question
  answer: Answer

class Answer:
  answer: str
```

Listing 3: Input and output type definitions for FinQA

```
class Rationale:
  steps: list[str]
```

Listing 4: `Rationale` type definition

```
class QuestionAnswer:
  question: Question
  answer: Answer
```

Listing 5: `QuestionAnswer` type definition

```
class ThinkingSteps:
  steps: list[str]

class Paragraph:
  title: str
  sentences: list[str]

class Context:
  paragraphs: list[Paragraph]

class SupportingFact:
  title: str
  sentence: str

class RelevantContext:
  sentences: list[str]

class Answer:
  answer: str

class Question:
  id: str
  question: str
  context: Context

class QuestionAnswer:
  question: Question
  answer: Answer

class AnswerFirstAttemptThinkingStepsAnswer:
  answer_first_attempt: Answer
  thinking_steps: ThinkingSteps
  answer: Answer

class QuestionAnswerFirstAttempt:
  question: Question
  answer_first_attempt: Answer

class
    QuestionAnswerFirstAttemptThinkingSteps:
  question: Question
  answer_first_attempt: Answer
```

```
  thinking_steps: ThinkingSteps
```

Listing 6: Type definitions for **refine-structure** on HotPotQA

```
class Expression:
  operator: Literal['+', '-', '*', '/']
  left: Union[int, 'Expression']
  right: Union[int, 'Expression']

class Answer:
  answer: Expression
```

Listing 7: `Expression` type definitions in MGSM **expression-cascade-structure** experiments,numbers=none

## G.2 DSPY SETUP

We conduct most of the DSPy experiments under v 3.0.1, but report results from DSPy v 2.6.19 for `gemini-1.1-7b-it` experiments since both BFSWRS and MIPROv2 struggle to generate valid outputs under DSPy v 3.0.1. Moreover, the non-optimized MGSM average accuracy is much lower under v 3.0.1 (for Native CoT it is 0.7% under v 2.6.19, and 0.2% under v 3.0.1). For all other experiments, we report results from DSPy v 3.0.1 which sets up JSON schema-based constrained decoding correctly out-of-the-box. As we noted in §4.2, constrained decoding significantly improves performance for tasks with structured output.

We serve base models on vLLM v 0.10.0.

**Input and output object definitions.** For structured input and output tasks, we subclass `dspy.Signature` as `QASignature` to represent examples. The property names and types in a `QASignature` class are identical to counterparts in TAC experiments. FinQA and MGSM-SymPy signatures are listed in Listing 8 and Listing 9 respectively.

```
class QASignature(dspy.Signature):
  pre_text: list[str] = dspy.InputField()
  table: list[list[str]] = dspy.InputField()
  post_text: list[str] = dspy.InputField()
  question: str = dspy.InputField()
  answer: str = dspy.OutputField()
```

Listing 8: DSPy object signature for FinQA. Property names and types are identical to their TAC counterparts in Listing 3

```
class Expression(pydantic.BaseModel):
  operator: Literal['+', '-', '*', '/']
  left: Union[int, float, 'Expression']
  right: Union[int, float, 'Expression']

class QASignature(dspy.Signature):
  question: str = dspy.InputField()
  answer: Expression = dspy.OutputField()
```

Listing 9: DSPy object signature for MGSM-SymPy. Property names and types are identical to their TAC counterparts in Listing 7

**DSPy models.** We conduct reasoning experiments on both the native `dspy.ChainOfThought` module, and an explicitly two-step composite module that resembles TAC **cot-cascade-structure** patterns. Two-step modules for FinQA and MuSR are listed in Listings 10 and 11 as examples.

```
class QuestionRationale(dspy.Signature):
  question: str = dspy.InputField()
  pre_text: list[str] = dspy.InputField()
```

```python
  table: list[list[str]] = dspy.InputField()
  post_text: list[str] = dspy.InputField()
  question: str = dspy.InputField()
  rationale: list[str] = dspy.OutputField()

class RationaleAnswer(dspy.Signature):
  rationale: list[str] = dspy.InputField()
  answer: str = dspy.OutputField()

class TwoStepPredictor(dspy.Module):
  def __init__(self):
    self.question_to_rationale = dspy.
        Predict(QuestionRationale)
    self.rationale_to_answer = dspy.Predict(
        RationaleAnswer)

  def forward(self, pre_text: list[str],
      table: list[list[str]], post_text:
      list[str], question: str):
    r = self.question_to_rationale(question=
        question, pre_text=pre_text, table=
        table, post_text=post_text).
        rationale
    return dspy.Prediction(answer=self.
        rationale_to_answer(rationale=r).
        answer)
```

Listing 10: DSPy two-step reasoning model definition for FinQA

```python
class QuestionRationale(dspy.Signature):
  context: str = dspy.InputField()
  question: str = dspy.InputField()
  choices: list[str] = dspy.InputField()
  rationale: list[str] = dspy.OutputField()

class RationaleAnswer(dspy.Signature):
  rationale: list[str] = dspy.InputField()
  choices: list[str] = dspy.InputField()
  answer: str = dspy.OutputField()

class TwoStepPredictor(dspy.Module):
  def __init__(self):
    self.question_to_rationale = dspy.
        Predict(QuestionRationale)
    self.rationale_to_answer = dspy.Predict(
        RationaleAnswer)

  def forward(self, context: str, question:
      str, choices: list[str]):
    r = self.question_to_rationale(question=
        question, context=context, choices=
        choices).rationale
    return dspy.Prediction(answer=self.
        rationale_to_answer(rationale=r,
        choices=choices).answer)
```

Listing 11: DSPy two-step reasoning model definition for MuSR

**Prompt optimization under DSPy.** We experiment with optimizers `dspy.MIPROv2` and `dspy.BootstrapFewShotWithRandomSearch` (listed as BFSWRS below). For MGSM-SymPy and FinQA experiments we do not report BFSWRS results, as they consistently need more context length than the model maximum (8192). Moreover, for FinQA experiments we resort to MIPROv2 0-shot due to similar context length problems.

We set `max_errors=2` for all optimizers. For MiPROv2 we set `auto='medium'`. For MiPROv2 with 0-shot settings we additionally set `max_bootstrapped_demos=0`, `max_labed_demos=0`.

## H  PER-LANGUAGE TAC AND ORIGINAL STaR MGSM AND MGSM-SYMPY RESULTS

Per-language TAC and original STaR experimental results on tasks MGSM and MGSM-SymPy are listed in Tables 6 and 7.

| Pattern | Adaptation Method | es | en | de | fr | zh | ru | ja | te | th | Average |
|---|---|---|---|---|---|---|---|---|---|---|---|
| direct | TACSTaR | 27.5 | 27.5 | 25.0 | 25.0 | 23.3 | 25.8 | 23.3 | 18.3 | 26.7 | 24.7 |
| cot-type-structure | TACSTaR | 80.0 | 84.2 | 76.7 | 83.3 | 80.0 | 85.0 | 71.7 | 79.2 | 83.3 | 80.4 |
| cot-cascade-structure | TACSTaR | 87.5 | 87.5 | 83.3 | 85.8 | 80.0 | 87.5 | 74.2 | 73.3 | 80.8 | 82.2 |
| refine-structure | TACSTaR | 86.7 | 90.0 | 76.7 | 77.5 | 73.3 | 78.3 | 69.2 | 72.5 | 83.3 | 78.6 |
| expression-cascade-structure | TACSTaR | 83.3 | 82.5 | 83.3 | 75.8 | 70.0 | 79.2 | 65.8 | 75.0 | 75.8 | 75.9 |
| cot-cascade-structure | un-adapted | 42.5 | 47.5 | 46.7 | 42.5 | 45.0 | 53.3 | 31.7 | 45.0 | 54.2 | 45.4 |
| cot-type-structure | un-adapted | 77.5 | 79.2 | 80.8 | 76.7 | 68.3 | 79.2 | 68.3 | 69.2 | 73.3 | 74.7 |
| expression-cascade-structure | un-adapted | 76.7 | 71.7 | 69.2 | 70.8 | 68.3 | 68.3 | 63.3 | 70.8 | 73.3 | 69.5 |
| cot-cascade-structure | amortized TACSTaR | 84.2 | 91.7 | 86.7 | 83.3 | 82.5 | 81.7 | 70.8 | 77.5 | 83.3 | 82.4 |
| N/A | original STaR | 74.2 | 79.2 | 75.8 | 75.8 | 70.0 | 88.3 | 74.2 | 75.8 | 75.8 | 76.9 |

Table 6: `gemma-2-27b-it` MGSM and MGSM-SymPy per-language accuracies (TAC and original STaR experiments).

| Pattern | Adaptation Method | es | en | de | fr | zh | ru | ja | te | th | Average |
|---|---|---|---|---|---|---|---|---|---|---|---|
| direct | TACSTaR | 5.8 | 6.7 | 6.7 | 8.3 | 7.5 | 2.5 | 5.0 | 1.7 | 1.7 | 5.1 |
| cot-cascade-structure | TACSTaR | 40.8 | 35.8 | 31.7 | 29.2 | 24.2 | 31.7 | 13.3 | 18.3 | 20.8 | 27.3 |
| cot-cascade-structure | un-adapted | $8.0 \cdot 10^{-1}$ | 0.0 | $8.0 \cdot 10^{-1}$ | 0.0 | 0.0 | $8.0 \cdot 10^{-1}$ | 0.0 | 1.7 | 0.0 | $5.0 \cdot 10^{-1}$ |
| N/A | original STaR | 15.0 | 27.5 | 1.7 | 5.8 | 22.5 | 0.0 | 3.3 | 9.2 | 9.2 | 10.5 |

Table 7: `gemma-1.1-7b-it` MGSM per-language accuracies (TAC and original STaR experiments).

## I  PER-TASK TAC MuSR RESULTS

Per-task TAC experimental results on task MuSR are listed in Tables 8 and 9.

| Decoding Method | Murder Mystery | Object Placements | Team Allocation | Average |
|---|---|---|---|---|
| Generation | 61.7 | 51.6 | 41.7 | 51.6 |
| Classification | 65.0 | 50.0 | 80.0 | 65.0 |

Table 8: `gemma-2-27b-it` MuSR per-task accuracies (TAC experiments).

| Decoding Method | Murder Mystery | Object Placements | Team Allocation | Average |
|---|---|---|---|---|
| Generation | 60.0 | 43.7 | 82.5 | 62.1 |
| Classification | 59.2 | 42.9 | 85.8 | 62.6 |

Table 9: `gemma-1.1-7b-it` MuSR per-task accuracies (TAC experiments).

## J  PER-TASK DSPY MuSR RESULTS

Per-task DSPy experimental results on task MuSR are listed in Tables 10 and 11.

## K  PER-LANGUAGE DSPY MGSM AND MGSM-SYMPY RESULTS

Per-language DSPy experimental results on tasks MGSM and MGSM-SymPy are listed in Tables 13 to 15.

| Model | Optimizer | Murder Mystery | Object Placements | Team Allocation | Average |
|-------|-----------|----------------|-------------------|-----------------|---------|
| Native CoT | None | 20.8 | 0.0 | 0.0 | 6.9 |
| Native CoT | MIPRO 0-shot | 40.8 | $7.9 \cdot 10^{-1}$ | 0.0 | 13.9 |
| Native CoT | MIPRO | 51.7 | 50.8 | 49.2 | 50.5 |
| Two-step | None | 52.5 | 14.3 | 22.5 | 29.8 |
| Two-step | MIPRO 0-shot | 55.0 | 27.8 | 19.2 | 34.0 |
| Two-step | MIPRO | 59.2 | 44.4 | 50.8 | 51.5 |

Table 10: `gemma-2-27b-it` MuSR per-task accuracies (DSPy experiments).

| Model | Optimizer | Murder Mystery | Object Placements | Team Allocation | Average |
|-------|-----------|----------------|-------------------|-----------------|---------|
| Native CoT | None | 10.0 | 3.2 | 3.3 | 5.5 |
| Native CoT | MIPRO 0-shot | 6.7 | 3.2 | 2.5 | 4.1 |
| Native CoT | MIPRO | 34.2 | 25.4 | 50.0 | 36.5 |
| Two-step | None | 33.3 | 5.6 | 16.7 | 18.5 |
| Two-step | MIPRO 0-shot | 35.8 | 1.6 | 15.0 | 17.5 |
| Two-step | MIPRO | 44.2 | 32.5 | 26.7 | 34.5 |

Table 11: `gemma-1.1-7b-it` MuSR per-task accuracies (DSPy experiments).

| Model | Optimizer | Murder Mystery | Object Placements | Team Allocation | Average |
|-------|-----------|----------------|-------------------|-----------------|---------|
| Native CoT | None | 0.0 | 0.0 | 0.0 | 0.0 |
| Native CoT | MIPRO 0-shot | 0.0 | 0.0 | 0.0 | 0.0 |
| Native CoT | MIPRO | 55.8 | 50.8 | 47.5 | 51.4 |
| Two-step | None | 4.2 | $7.9 \cdot 10^{-1}$ | 0.0 | 1.7 |
| Two-step | MIPRO 0-shot | 3.3 | 1.6 | 0.0 | 1.6 |
| Two-step | MIPRO | 65.0 | 59.5 | 60.0 | 61.5 |

Table 12: `Qwen3-8B` MuSR per-task accuracies (DSPy experiments).

| Model | Optimizer | es | en | de | fr | zh | ru | ja | te | th | Average |
|-------|-----------|-----|-----|-----|-----|-----|-----|-----|-----|-----|---------|
| Native CoT | None | 55.0 | 57.5 | 52.5 | 51.7 | 54.2 | 59.2 | 45.0 | 39.2 | 40.0 | 50.5 |
| Native CoT | BFSWRS | 84.2 | 89.2 | 87.5 | 81.7 | 75.0 | 87.5 | 75.0 | 77.5 | 79.2 | 81.9 |
| Native CoT | MIPROv2 | 82.5 | 86.7 | 81.7 | 76.7 | 77.5 | 84.2 | 70.0 | 74.2 | 75.8 | 78.8 |
| Two-step | None | 1.7 | 5.8 | 2.5 | 1.7 | 3.3 | 1.7 | 1.7 | 3.3 | 5.0 | 3.0 |
| Two-step | MIPROv2 | 76.7 | 83.3 | 76.7 | 78.3 | 73.3 | 79.2 | 70.0 | 67.5 | 71.7 | 75.2 |
| Two-step | BFSWRS | 80.8 | 84.2 | 76.7 | 81.7 | 70.0 | 81.7 | 67.5 | 64.2 | 72.5 | 75.5 |

Table 13: `gemma-2-27b-it` MGSM per-language accuracies (DSPy experiments).

| Model | Optimizer | es | en | de | fr | zh | ru | ja | te | th | Average |
|-------|-----------|-----|-----|-----|-----|-----|-----|-----|-----|-----|---------|
| Native CoT | None | $8.0 \cdot 10^{-1}$ | $8.0 \cdot 10^{-1}$ | $8.0 \cdot 10^{-1}$ | 0.0 | 0.0 | 2.5 | 1.7 | 0.0 | 0.0 | $7.0 \cdot 10^{-1}$ |
| Native CoT | BFSWRS | 0.0 | $8.0 \cdot 10^{-1}$ | 1.7 | 5.0 | $8.0 \cdot 10^{-1}$ | 1.7 | 1.7 | 2.5 | 0.0 | 1.6 |
| Native CoT | MIPROv2 | $8.0 \cdot 10^{-1}$ | 1.7 | 2.5 | 2.5 | 1.7 | 0.0 | 1.7 | $8.0 \cdot 10^{-1}$ | $8.0 \cdot 10^{-1}$ | 1.4 |
| Two-step | None | 0.0 | 0.0 | $8.0 \cdot 10^{-1}$ | 0.0 | 0.0 | 0.0 | 1.7 | 0.0 | 0.0 | $3.0 \cdot 10^{-1}$ |
| Two-step | MIPROv2 | 0.0 | 0.0 | 0.0 | 0.0 | 0.0 | $8.0 \cdot 10^{-1}$ | 0.0 | 0.0 | $8.0 \cdot 10^{-1}$ | $2.0 \cdot 10^{-1}$ |
| Two-step | BFSWRS | 0.0 | 0.0 | 0.0 | 0.0 | 0.0 | $8.0 \cdot 10^{-1}$ | 0.0 | 0.0 | $8.0 \cdot 10^{-1}$ | $2.0 \cdot 10^{-1}$ |

Table 14: `gemma-1.1-7b-it` MGSM per-language accuracies (DSPy experiments).

## L  DSPY FINQA RESULTS

DSPy experimental results on the FinQA task are listed in Table 16 and Table 17.

| Model | Optimizer | es | en | de | fr | zh | ru | ja | te | th | Average |
|-------|-----------|-----|-----|-----|-----|-----|-----|-----|-----|-----|---------|
| Native CoT | None | 56.7 | 66.7 | 55.0 | 45.8 | 47.5 | 59.2 | 45.0 | 49.2 | 45.8 | 52.3 |
| Native CoT | MIPROv2 | 66.7 | 64.2 | 58.3 | 60.8 | 56.7 | 62.5 | 50.8 | 42.5 | 51.7 | 57.1 |
| Two-step | None | 0.0 | 0.0 | 0.0 | 0.0 | 0.0 | 0.0 | 0.0 | 0.0 | 0.0 | 0.0 |
| Two-step | MIPROv2 | 0.0 | 0.0 | 0.0 | 0.0 | 0.0 | 0.0 | 0.0 | 0.0 | 0.0 | 0.0 |

Table 15: `gemma-2-27b-it` MGSM-SymPy per-language accuracies (DSPy experiments).

| Model | Optimizer | Accuracy |
|-------|-----------|----------|
| Native CoT | None | 11.7 |
| Native CoT | MIPROv2 0-shot | 12.7 |
| Two-step | None | 5.7 |
| Two-step | MIPROv2 0-shot | 10.7 |

Table 16: `gemma-2-27b-it` FinQA accuracy (DSPy experiments).

| Model | Optimizer | Accuracy |
|-------|-----------|----------|
| Native CoT | None | 0.0 |
| Native CoT | MIPROv2 0-shot | $6.7 \cdot 10^{-1}$ |
| Two-step | None | 0.0 |
| Two-step | MIPROv2 0-shot | $3.3 \cdot 10^{-1}$ |

Table 17: `gemma-1.1-7b-it` FinQA accuracy (DSPy experiments).

| Model | Optimizer | Accuracy |
|-------|-----------|----------|
| Native CoT | None | 4.3 |
| Native CoT | MIPROv2 0-shot | 5.3 |
| Two-step | None | 1.0 |
| Two-step | MIPROv2 0-shot | 12.0 |

Table 18: `Qwen3-8B` FinQA accuracy (DSPy experiments).

## M  EXAMPLE EXPRESSIONS FROM **expression-cascade-structure** UNDER THE MGSM-SYMPY TASK

See Table 19.

| Question | Answer | Expression |
|----------|--------|------------|
| Nissa hires 60 seasonal workers to play elves in her department store's Santa village. A third of the elves quit after children vomit on them, then 10 of the remaining elves quit after kids kick their shins. How many elves are left? | 20 | $(60 - (60/3)) - 10$ |
| The expenditure of Joseph in May was \$500. In June, his expenditure was \$60 less. How much was his total expenditure for those two months? | 940 | $500 + 440$ |
| Tom gets 4 car washes a month. If each car wash costs \$15 how much does he pay in a year? | 720 | $(15 \times 4) \times 12$ |

Table 19: Example arithmetic expressions generated for MGSM questions by **expression-cascade-structure**.

## N   EXAMPLE INSTRUCTION PROMPT GENERATED BY LANGFUN

The Langfun library translates requests that transformed a typed object into another typed object into natural language instructions for LLMs, to facilitate its `parse` operations. For example, Listing 12 is a prompt generated by Langfun for the request that transforms a `Question` object into an Answer object.

```
Please respond to the last INPUT_OBJECT with
    OUTPUT_OBJECT according to OUTPUT_TYPE.

INPUT_OBJECT:
  1 + 1 =

OUTPUT_TYPE:
  Answer

  ```python
  class Answer:
    final_answer: int
  ```

OUTPUT_OBJECT:
  ```python
  Answer(
    final_answer=2
  )
  ```

INPUT_OBJECT:
  ```python
  Question(
    question='How are you?'
  )
  ```

OUTPUT_TYPE:
  Answer

  ```python
  class Answer:
    answer: str
  ```

OUTPUT_OBJECT:
```

Listing 12: Example instruction prompt generated by Langfun

## O   PERFORMANCE-COMPUTATION COST ANALYSIS

We conduct additional experiments on FinQA (Gemma 2 27B) to analyze the cost-benefit trade-off, using token counts (*i.e.*, tokens used during training, and tokens used for evaluation on the entire test set) as a proxy for cost. On this task, the TACSTaR run processed $\sim 27M$ training tokens. The baseline DSPy optimization (MIPROv2, using the default `auto='medium'` configuration) processed $\sim 1.6M$ tokens. Decoding the 300 test examples takes $\sim 29M$ tokens for TAC (using 32 samples for robust estimation as described in §G.1) and $\sim 0.5M$ tokens for DSPy.

We evaluated whether increasing DSPy's compute budget closes the performance gap:

- **Scaling Training**: We increased the `num_trials` hyperparameter under MIPRO v2 from 12 (under the `auto='medium'` default setting) to 32 ($\sim 3.7M$ tokens) and 300 ($\sim 33M$ tokens).

- **Scaling Inference**: We used majority voting (with the same T=1.0, top_p=0.95, top_k=40 settings) with ensemble sizes 100 ($\sim 55M$ tokens) and 500 ($\sim 260M$ tokens).

Results listed in Table 20 show that DSPy performance plateaus quickly. Even when significantly increasing DSPy's training and test budget (up to 9x TAC's inference cost), the accuracy (14.3) remains far below TACs (34.0).

| Training budget | Test budget | Training token count | | | Test token count | | | Accuracy |
|---|---|---|---|---|---|---|---|---|
| | | Encoded | Decoded | Total | Encoded | Decoded | Total | |
| 12 | 1 | 1577489 | 107297 | 1684786 | 489646 | 35302 | 524948 | 12.7 |
| 12 | 100 | 1362700 | 98030 | 1460730 | 50714928 | 4695123 | 55410051 | 12.7 |
| 12 | 500 | 1408479 | 95815 | 1504294 | 243050795 | 18271389 | 261322184 | 12.3 |
| 32 | 100 | 3246101 | 226967 | 3473068 | 50074600 | 4747957 | 54822557 | 12.0 |
| 32 | 500 | 3429654 | 240376 | 3670030 | 247409788 | 20255592 | 267665380 | 14.3 |
| 300 | 100 | 31112328 | 2146258 | 33258586 | 48754600 | 3655251 | 52409851 | 12.3 |
| 300 | 500 | 31216543 | 2164159 | 33380702 | 246926388 | 23154727 | 270081115 | 12.7 |
| TAC | | 26007546 | 1349250 | 27356796 | 27517698 | 1476911 | 28994609 | 34.0 |

Table 20: Cost analysis. Training budget corresponds to the `num_trials` hyperparameter under MIPRO v2, and test budget corresponds to ensemble size. Accuracy numbers from the first and last rows are copied from Table 16 and Fig. 2a respectively.

## P    SNIS QUALITY OF LEARNED PROPOSAL DISTRIBUTIONS

Specifically, we rewrite $\mathrm{KL}[\tilde{q}_{\phi}||p_{\theta}] - \mathrm{KL}[\tilde{p}_{\text{fallback}}||p_{\theta}]$ as:

$$\mathrm{KL}[\tilde{q}_{\phi}||p_{\theta}] - \mathrm{KL}[\tilde{p}_{\text{fallback}}||p_{\theta}] = \mathbb{E}_{z \sim \tilde{q}_{\phi}(\cdot|z_1^*, z_2^*)}[\log \tilde{q}_{\phi}(z \mid z_1^*, z_2^*) - \log p_{\theta}(z \mid z_1^*, z_2^*)]$$
$$- \mathbb{E}_{z \sim \tilde{p}_{\text{fallback}}(\cdot|z_1^*, z_2^*)}[\log \tilde{p}_{\text{fallback}}(z \mid z_1^*, z_2^*) - \log p_{\theta}(z \mid z_1^*, z_2^*)]$$
$$= \underbrace{\mathbb{E}_{z \sim \tilde{q}_{\phi}(\cdot|z_1^*, z_2^*)}[\log \tilde{q}_{\phi}(z \mid z_1^*, z_2^*) - \log p_{\theta}(z, z_2 = z_2^* \mid z_1^*)]}_{\text{call this } \mathrm{K}_{\tilde{q}_{\phi}}}$$
$$- \underbrace{\mathbb{E}_{z \sim \tilde{p}_{\text{fallback}}(\cdot|z_1^*, z_2^*)}[\log \tilde{p}_{\text{fallback}}(z \mid z_1^*, z_2^*) - \log p_{\theta}(z, z_2 = z_2^* \mid z_1^*)]}_{\text{call this } \mathrm{K}_{\tilde{p}_{\text{fallback}}}},$$

(13)

where we exploit the identity

$$\mathbb{E}_{q_1}[\log p_{\theta}(z \mid z_1^*, z_2^*)] - \mathbb{E}_{q_2}[\log p_{\theta}(z \mid z_1^*, z_2^*)]$$
$$= \mathbb{E}_{q_1}[\log p_{\theta}(z, z_2 = z_2^* \mid z_1^*) - \underbrace{\log\left(\sum_{z'} p_{\theta}(z^*, z_2 = z_2^* \mid z_1^*)\right)}_{=C}]$$
$$- \mathbb{E}_{q_2}[\log p_{\theta}(z \mid z_1^*, z_2^*) - \log\left(\sum_{z'} p_{\theta}(z^*, z_2 = z_2^* \mid z_1^*)\right)]$$
$$= \mathbb{E}_{q_1}[\log p_{\theta}(z, z_2 = z_2^* \mid z_1^*)] - \mathbb{E}_{q_2}[\log p_{\theta}(z, z_2 = z_2^* \mid z_1^*)] - \mathbb{E}_{q_1}[C] + \mathbb{E}_{q_2}[C]$$
$$= \mathbb{E}_{q_1}[\log p_{\theta}(z, z_2 = z_2^* \mid z_1^*)] - \mathbb{E}_{q_2}[\log p_{\theta}(z, z_2 = z_2^* \mid z_1^*)].$$

To compute $\mathrm{KL}[\tilde{q}_{\phi}||p_{\theta}] - \mathrm{KL}[\tilde{p}_{\text{fallback}}||p_{\theta}]$, we compute the two expectations $\mathrm{K}_{\tilde{q}_{\phi}}$ and $\mathrm{K}_{\tilde{p}_{\text{fallback}}}$ in Eq. (13), constructing the self-normalized distribution $\tilde{p}_{\text{fallback}}(z \mid z_1^*, z_2^*) \propto \frac{p_{\theta}(z, z_2 = z_2^*|z_1^*)}{p_{\text{fallback}}(z|z_1^*, z_2^*)}$ with 16 samples drawn from $p_{\text{fallback}}$. We also construct $\tilde{q}_{\phi}(z \mid z_1^*, z_2^*)$ this way with 16 samples.

## Q    EFFECTIVE SAMPLE SIZE OF TRAINED MODELS

We conducted additional experiments on MGSM (Gemma 7B) to analyze whether diversity collapses as type compliance increases during training (§4.4). We use Effective Sample Size (ESS) normalized

between $0$ and $1$ as our evaluation metric. We measure ESS (using $16$ samples for each validation example) at the end of each training epoch on the validation datasets. Results are listed in Fig. 9. We find the average ESS fluctuates between $0.4$ to $0.5$, implying that the samples did not collapse to the mode (in which case normalized ESS would be around $1/16 = 0.625$).

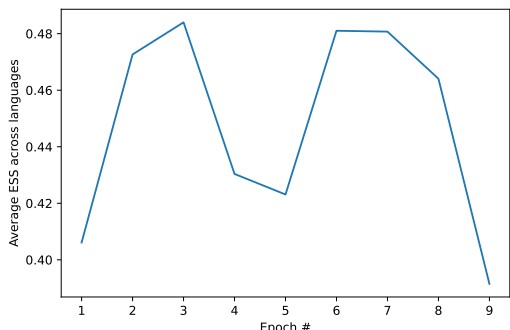

Figure 9: Average ESS at end of epochs. Sample size $= 16$. ESS value normalized $\in [\frac{1}{16}, 1]$ in plot.

## R  ESTIMATING MARGINAL PROBABILITIES VIA IMPORTANCE SAMPLING

A key advantage of the probabilistic framing of TACs is the ability to estimate marginal probabilities, even when the distribution is unnormalized. LM adaptors in a TAC can be used as proposal distributions for importance sampling.

Let $\mathbf{z}_m$ be a node coming out of an LM adaptor. An $N$-sample estimate of the unnormalized probability that $\mathbf{z}_m$ equals $c$, conditioned on the input $\mathbf{z}_1$, $\tilde{p}(\mathbf{z}_m = c \mid \mathbf{z}_1; \boldsymbol{\theta})$, is:

$$\hat{\tilde{p}}_{|\mathbf{z}_1}(m, c, N) = \sum_{n=1}^{N} \left[ \frac{p_{LM}(\mathbf{z}_m = c; \boldsymbol{\theta})}{N \cdot p_{LM}(\mathbf{z}_m = \mathbf{z}_m^{(n)}; \boldsymbol{\theta})} \right] \tag{14}$$

where $\mathbf{z}_m^{(n)}$ is the $n$-th sample of $\mathbf{z}_m$ (drawn using Algorithm 1). Equation (14) is an unbiased importance sampling estimate of the unnormalized probability $\tilde{p}(\mathbf{z}_m = c \mid \mathbf{z}_1; \boldsymbol{\theta})$ (since $\mathrm{supp}(\tilde{p}) \subseteq \mathrm{supp}(p_{LM})$).

In the special case that $\mathbf{z}_m$ has finite support (e.g., a classification task), we can estimate the *normalized* marginal probability using Self-Normalized Importance Sampling (SNIS) by renormalizing over all possible values $c'$:

$$\hat{p}(\mathbf{z}_m = c \mid \mathbf{z}_1; \boldsymbol{\theta}) = \frac{\hat{\tilde{p}}_{|\mathbf{z}_1}(m, c, N)}{\sum_{c'} \hat{\tilde{p}}_{|\mathbf{z}_1}(m, c', N)}. \tag{15}$$

